# The actin cytoskeletal architecture of estrogen receptor positive breast cancer cells suppresses invasion

Marco Padilla-Rodriguez[1], Sara S. Parker [1], Deanna G. Adams[1], Thomas Westerling[2], Julieann I. Puleo[1], Adam W. Watson[1], Samantha M. Hill[1], Muhammad Noon[3], Raphael Gaudin [4,5], Jesse Aaron [6], Daoqin Tong[7], Denise J. Roe[8], Beatrice Knudsen[9] & Ghassan Mouneimne[1]

Estrogen promotes growth of estrogen receptor-positive (ER+) breast tumors. However, epidemiological studies examining the prognostic characteristics of breast cancer in post-menopausal women receiving hormone replacement therapy reveal a significant decrease in tumor dissemination, suggesting that estrogen has potential protective effects against cancer cell invasion. Here, we show that estrogen suppresses invasion of ER+ breast cancer cells by increasing transcription of the Ena/VASP protein, EVL, which promotes the generation of suppressive cortical actin bundles that inhibit motility dynamics, and is crucial for the ER-mediated suppression of invasion in vitro and in vivo. Interestingly, despite its benefits in suppressing tumor growth, anti-estrogenic endocrine therapy decreases EVL expression and increases local invasion in patients. Our results highlight the dichotomous effects of estrogen on tumor progression and suggest that, in contrast to its established role in promoting growth of ER+ tumors, estrogen has a significant role in suppressing invasion through actin cytoskeletal remodeling.

---

[1] Department of Cellular and Molecular Medicine, University of Arizona Cancer Center, University of Arizona, Tucson 85724 AZ, USA. [2] Department of Medical Oncology, Division of Molecular and Cellular Oncology, and Center for Functional Cancer Epigenetics, Dana-Farber Cancer Institute and Harvard Medical School, Boston 02115 MA, USA. [3] BioComputing Facility, University of Arizona, Tucson 85724 AZ, USA. [4] INSERM U1110, Institut de Recherche sur les Maladies Virales et Hépatiques, Strasbourg 67000, France. [5] Université de Strasbourg, Strasbourg 67000, France. [6] Advanced Imaging Center, Howard Hughes Medical Institute, Janelia Research Campus, Ashburn 20147 VA, USA. [7] School of Geography and Development, College of Social and Behavioral Sciences, University of Arizona, Tucson 85724 AZ, USA. [8] University of Arizona Cancer Center and Mel and Enid Zuckerman College of Public Health, University of Arizona, Tucson 85724 AZ, USA. [9] Department of Biomedical Sciences, Cedars-Sinai Medical Center, Los Angeles 90048 CA, USA. Correspondence and requests for materials should be addressed to G.M. (email: gmouneimne@email.arizona.edu)

Estrogen receptor-positive (ER+) breast cancers are the most commonly diagnosed subgroup of breast tumors, and most breast cancer deaths are caused by metastatic ER+ tumors[1,2]. Several lines of evidence suggest that the risk of ER+ breast cancer increases with estrogen exposure during a women's lifetime, for example, due to earlier menarche or late menopause (i.e., longer exposure to reproductive hormones due to longer ovarian activity)[3]. Moreover, large-scale clinical trials designed to look at the effects of hormone replacement therapy (HRT) on breast cancer incidence in postmenopausal women revealed that HRT increased the risk of breast cancer[4,5]. However, extended exposure to estrogen during HRT was associated with less dissemination and better outcome[5]. Interestingly, HRT did not reduce the locoregional recurrence rate[6], suggesting that under HRT, recurrent tumors are able to develop and grow locally at the initial tumor site but are less prone to disseminate and metastasize to distant sites.

In this study, we investigated this potential protective role of estrogen against cancer dissemination and metastasis. In a meta-analysis, including 17,497 patients from 10 clinical cross-sectional studies, we found that the metastatic burden in patients who developed breast cancer while on estrogen treatment was reduced. In addition, we found that ER is associated with lower invasive capacity. Despite the significant role of actin remodeling in cell invasion, the hormonal regulation of the actin cytoskeletal architecture in ER+ breast cancer cells, is not known. We found that ER promotes the formation of distinct actin structures with protective properties against invasion. We used a multimodal targeted discovery approach to examine the transcriptional regulation of actin cytoskeletal regulators by ER. Among a comprehensive list of known actin regulators, we identified a member of the Ena/VASP family of proteins, *EVL*, as the most significant transcriptional target of ER. We found that EVL is up-regulated in ER+ tumors and suppresses invasion, and that EVL levels are reduced in tumors after anti-estrogenic hormone therapy, which was associated with increased invasion.

## Results

**ER is associated with low breast cancer dissemination**. In a meta-analysis of ten epidemiological studies[6–15], we investigated lymph node (LN) positivity in postmenopausal women diagnosed with breast cancer while on HRT (current-users), compared to women who never used HRT (never-users). We used LN status as a metric of tumor cell dissemination and the commonly used cutoff of >3 (LN >3) as a marker of distant metastasis beyond the regional axillary LNs. HRT was associated with lower odds ratio (OR) of having LN >3, suggesting less distant dissemination (Fig. 1a, b). Moreover, to examine both regional and distant dissemination, we performed a binary assessment of LN status (LN+ or LN−). This analysis showed that OR of having LN+ was lower in HRT current-users, indicating less regional and distant dissemination (Supplementary Fig. 1a, b). In addition, in ER+ breast cancer, tumors with low LN dissemination had higher ER levels, as compared to tumors with high LN dissemination (1.8-fold in LN ≤3 and 1.5-fold in LN−, compared to LN >3 and LN+, respectively); not surprisingly, patients with less LN-disseminated tumors have higher survival rate (Supplementary Fig. 1c, d).

We investigated the effect of ER on cancer cell invasion, the initial step in metastatic dissemination, in breast cancer patient samples from two tissue microarrays (TMA#1 and TMA#2; see Methods). We determined the local invasion index (LII) for each tumor sample by employing the nearest neighbor distance (NND) approach, typically used in spatial analysis to study the second-order effect or local variation of point patterns[16]. Treating cancer

cells (identified by cytokeratin positivity) as stochastic events in a point pattern analysis, we measured the distance between the nucleus of each cell and the nucleus of its most proximal neighboring cell (NND) and calculated LII as the average NND within each tumor sample (Fig. 1c, d).

In TMA#1, we analyzed 64 samples from ER+ and ER− tumors, exhibiting a wide range of LN dissemination. Not surprisingly, compared to the more aggressive ER− tumors, ER+ tumors exhibited significantly lower LII (Fig. 1e, f), suggesting that ER positivity is associated with lower invasion rate. Moreover, tumors with LII ≤7 μm (denoted as LII low) were associated with foldfold less LN dissemination than tumors with LII ≥9 μm (denoted as LII high) (Supplementary Fig. 1e), suggesting that, in addition to ER positivity, low local invasion corresponds, as expected, to low distant LN dissemination. These data suggest that local invasion is a valid parameter to assess dissemination.

Furthermore, we investigated the association between ER expression and invasion in TMA#2, which comprised samples from ER+ luminal B tumors. Among ER+ tumors, luminal B tumors disseminate more extensively and exhibit a wide range of ER expression levels[17]. Quantitative analysis of ER levels (average intensity of ER immunofluorescence in each tumor sample) and LII in 180 samples from TMA#2 revealed a significant negative linear correlation between ER expression and invasion (Fig. 1g–i). In addition to expressing higher ER, low LII tumors are associated with less LN dissemination as compared to high LII tumors (Supplementary Fig. 1f). Moreover, analysis of survival rates in a cohort of patients with luminal B tumors[18] shows that high ER is associated with better outcome (Supplementary Fig. 1g). Together, these data demonstrate that in luminal B breast cancer, tumors with high ER expression exhibit low dissemination and are less invasive.

To investigate the direct effect of altering ER activity on invasion, we developed an in vitro culture system that allowed us to quantify the level of invasion of ER+ MCF7 breast cancer cells into 3D matrix under estradiol (E2) or ER inhibitor treatment (Fig. 1j). We found that ER inhibition enhanced invasion of MCF7 cells, whereas E2 suppressed it (Fig. 1k, l). These results are consistent with previous reports of ER-suppressing invasiveness of breast cancer cells[19,20]. Conversely, as expected, ER inhibition suppressed proliferation of the cells, whereas E2 promoted proliferation (Supplementary Fig. 1h). Together, these results suggest that the observed changes in invasion upon altering ER activity are not the result of changes in the proliferation rate, but are rather induced by enhanced cell motility.

**ER promotes suppressive cortical actin bundles**. Using kymography analysis, we quantified the motility dynamics of membrane protrusion, which is crucial for cancer cell invasion[21]. This analysis showed a marked decrease in the number and speed of membrane ruffles in E2-treated MCF7 cells, as compared to control (Fig. 2a–c and Supplementary Movie 1). Conversely, protrusive activity was significantly enhanced at the leading edge of MCF7 cells treated with ER inhibitors (Fig. 2a–c). We validated these data in ER+ breast cancer T47D cells (Fig. 2d–f and Supplementary Movie 2). Mechanistically, enhanced motility dynamics are well-established characteristics of an aggressive invasive behavior[22]; therefore, the suppression of protrusive activity by ER is in line with the low dissemination of tumor cells with high ER expression.

Motility dynamics of membrane protrusions are regulated by the architecture of the actin cytoskeleton[22]. Examining actin cytoskeletal remodeling by ER, we found that E2-treated MCF7 cells exhibited prominent cortical actin bundles, which were

concurrent with the absence of protrusions. Conversely, ER inhibition diminished these bundles and enhanced cell scattering and dissemination (Supplementary Fig. 2a, b and Supplementary Movie 3). Similar actin bundles have been previously observed in normal cells, where they constitute a contractile cortical barrier that suppresses Arp2/3-mediated protrusions[23,24]. We

quantitated the levels of phosphorylated myosin light chain (pMLC), as a contractility marker[25], and of Arp2/3 at the leading edge after altering ER activity in ER+ breast cancer cells. Considering the 3D morphometric properties of cortical bundles and protrusive ruffles, we used volumetric analysis of fluorescence intensity in confocal z-series (Fig. 2g). E2 treatment significantly

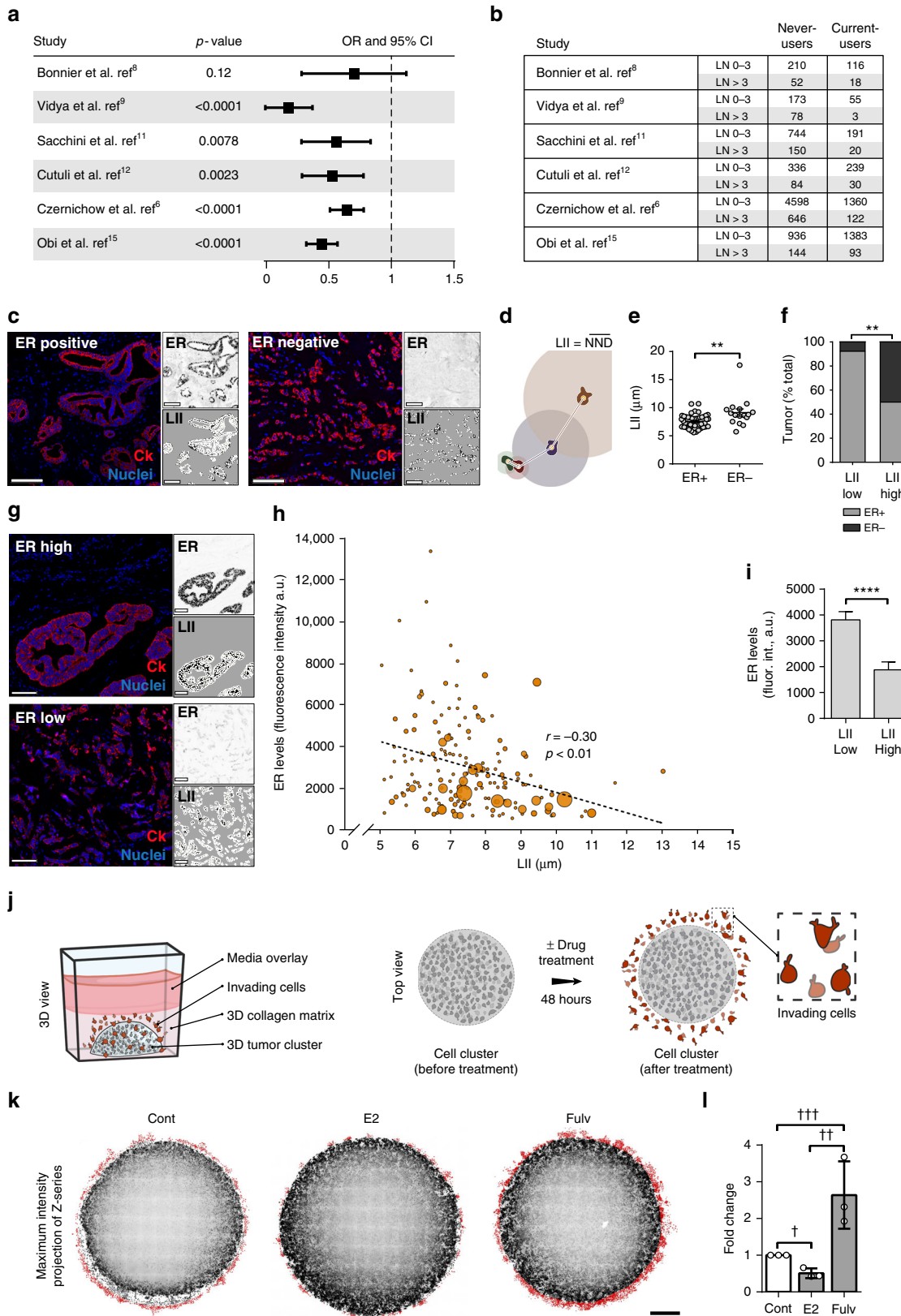

increased pMLC at cortical bundles, demonstrating elevated contractility, which was suppressed by ER inhibition (Fig. 2h, i and Supplementary Movie 4). The decrease in cortical contractility coincided with Arp2/3-positive leading edge ruffles; however, no significant changes in Arp2/3 levels at the leading edge were detected, suggesting that other actin regulators could be driving protrusion under ER inhibition (Fig. 2i). We confirmed these data in T47D cells (Fig. 2j, k). These results indicate that the suppressive effects of ER are mediated through remodeling of the actin cytoskeleton to generate suppressive cortical actin bunds (SCABs).

**SCABs suppress leading edge motility**. We further examined the relationship between cortical contractility and membrane motility dynamics. Using live-cell TIRFM, we imaged SCABs at the leading edge of MCF7 and T47D cells, before and after inhibiting contractility using the Rho-associated protein kinase (ROCK) inhibitor, Y-27632; the actin cytoskeleton was labeled with iRFP-Lifeact, and SCABs with MLC-mRuby2 (Supplementary Movie 5). Inhibiting contractility resulted in the loss of MLC at the leading edge and the dissolution of SCABs, and led to a rapid increase in protrusive activity, revealing a direct relationship between SCABs and the suppression of membrane motility dynamics (Fig. 3a, b). In addition, we validated the existence of SCABs, their contractile nature, and their suppressive effects in two types of epithelial cells: Madin-Darby canine kidney (MDCK) epithelial cells and Caco-2 human intestinal epithelial cells (Fig. 3c, d). Immunolabeling of pMLC in these cells revealed the contractile nature of these cortical structures, which corresponded with the absence of membrane protrusions (Fig. 3b). Importantly, the dissolution of SCABs dramatically increased the motility dynamics of the leading edge in both epithelial cell types (Fig. 3c). These data suggest that SCABs—and the attenuation of protrusive activity by SCABs—are present not only in ER+ breast cancer cells, but are common features of epithelial cells. ER is potentially enriching for SCABs by inducing the expression of actin regulators that promote SCAB formation.

**EVL is a transcriptional target of ER**. Remodeling of the actin cytoskeleton is controlled by a multitude of actin regulators whose differential expression in cancer leads to distinct architectures that impact invasion[26]. To identify actin-binding proteins that are transcriptionally regulated by ER, we took a targeted discovery approach by examining the differential expression of a curated list of 285 actin regulators in ER+ vs. ER− tumors using 12 datasets[18,27–37] (Supplementary Data 1, 2). The list of actin regulators was curated by first generating an extended list from

the Gene Ontology (GO) Consortium using the broad search term "actin cytoskeleton," and then refining it by limiting the selection to genes that specifically express actin-binding proteins. In examining 12 independent microarray datasets, the Ena/VASP family member, EVL, was most differentially expressed in ER+ tumors, compared to ER− tumors (Fig. 4a, Supplementary Fig. 3a, and Supplementary Data 2, 3). This result was validated using the breast cancer The Cancer Genome Atlas (TCGA) RNA-sequencing (RNA-seq) dataset (Supplementary Fig. 3b and Supplementary Data 4, 5). Moreover, unsupervised clustering analysis of the RNA-seq data showed that EVL closely clustered with ESR1, the gene encoding ERα (Fig. 4b), and their expression exhibited the highest correlation index (Supplementary Data 6).

Strikingly, using ER chromatin immunoprecipitation-sequencing (ChIP-seq) analysis, we found 12 high confidence ER peaks (defined in Methods) within the EVL gene and within 60 kb upstream of the transcription start site, suggesting a direct regulation of EVL by E2-stimulated ER (Fig. 4c). We further investigated four of the high confidence peaks by ChIP-quantitative real-time PCR (qPCR) and two peaks showed significant enrichment in ER binding after E2 stimulation as compared to negative control targets (Fig. 4d and Supplementary Data 7). Importantly, EVL mRNA expression was induced by E2 in both MCF7 and T47D cells and was suppressed by ER inhibition (Fig. 4e and Supplementary Fig. 3c). In addition, using immunofluorescence labeling, we observed similar changes in EVL protein levels in MCF7 cells after altering ER activity with the different treatments (Fig. 4f, g). Together, these results demonstrate that EVL is a bona fide transcriptional target of ER. Notably, EVL is a characterized suppressor of breast cancer cell invasion[26], making it an ideal candidate for mediating the protective effects of ER against invasion.

**EVL promotes ER-mediated actin remodeling**. EVL knockdown (KD), using validated EVL-targeting short hairpin (shRNA)[26] (Supplementary Fig. 4a, b), rendered cells unresponsive to E2 by blocking the generation of SCABs and preventing the expected suppression of membrane motility dynamics by E2 treatment (Fig. 5a, b, Supplementary Fig. 4c–e, and Supplementary Movie 6). Conversely, overexpression of eGFP-EVL restored the generation of SCABs and suppressed protrusion under ER inhibition (Fig. 5c, d, Supplementary Fig. 5a–e, and Supplementary Movie 7). These results indicate that EVL is necessary and sufficient to generate ER-mediated SCABs and promote the suppressive effects of ER on membrane motility dynamics.

Both overexpressed eGFP-EVL and endogenous EVL (visualized in CRISPR-tagged eGFP-EVL cells and in EVL-

**Fig. 1** ER expression is associated with low dissemination of breast cancer cells. **a** Meta-analysis of LN dissemination. Forest plot showing odds ratio, with 95% confidence interval (OR, 95% CI) of LN >3 in current-users compared with never-users of HRT. **b** Number of never-users and current-users with either LN ≤3 or LN >3 in the studies analyzed in **a**. **c** Representative images of ER+ (left panel) and ER− (right panel) tumors from TMA#1 (CDP-BCP-TMA), labeled for cytokeratin (red) and nuclei (blue). Scale bar is 100 μm. Top-right inset shows ER labeling and bottom-right inset shows binary masks of cytokeratin stain (black) and nuclei (orange). **d** Illustration of local invasion index (LII) measurement. Connecting lines represent the nearest neighbor distance (NND) of illustrated cells (color of each line matches color of corresponding cell). LII is calculated as mean NND. **e** Quantification of LII in TMA#1 (mean ± s.e.m); **p = 0.001 (unpaired t test). **f** Percentage of ER+ (gray) and ER− (black) tumors in low (≤7 μm) and high (≥9 μm) LII bins in TMA#1; **p = 0.002 (unpaired t test). **g** Representative images of luminal B breast tumors from TMA#2 (Cedars-Sinai LumB TMA) with high (top panel) or low (bottom panel) ER expression. Top-right inset shows ER labeling and bottom-right inset shows binary masks of cytokeratin stain (black) and nuclei (orange). Scale bar is 100 μm. **h** Scatter plot of LII and ER levels in TMA#2. For each data point, bubble area is proportional to the number of positive lymph nodes in the corresponding patient; r is Pearson's correlation coefficient; correlation is significant at p < 0.01. a.u. = aribitrary units. **i** ER levels in tumors with low (≤7 μm) and high (≥9 μm) LII in TMA#2; mean ± s.e.m. ****p < 0.0001 (Welch's t test). **j** Illustration of 3D culture system for quantification of invasion in vitro. Cells embedded in central area invade into surrounding collagen matrix. Zoomed-in illustration of the boxed area shows invading cells in red. **k** Maximum intensity projections of confocal z-series of ER+ breast cancer MCF7 cells treated with drug vehicle, estradiol (E2), or fulvestrant (fulv). Binary mask (red) highlights invaded cells. Scale bar is 500 μm. **l** Quantification of invasion. Data are from three independent experiments; mean ± s.d. †p = 0.003, ††p = 0.01, †††p = 0.03 (unpaired t test)

immunolabeled control cells) localized at SCABs (Fig. 5c and Supplementary Fig. 6a, b). In addition, EVL was localized at SCABs in cell clusters embedded in 3D collagen matrix (Supplementary Fig. 6c). To further analyze EVL localization at SCABs, we imaged mEos2-EVL and actin using interferometric photoactivated localization microscopy (iPALM), which allows for super-resolution molecular localization (within 10 nm) in the lateral and axial dimensions[38]. This analysis revealed that EVL enveloped the actin bundles in all dimensions at SCABs, as

discerned in orthogonal projections of cross-sectional segments taken across different regions of the SCABs (Fig. 5e and Supplementary Fig. 6d). Moreover, EVL was highly enriched at the tips of SCABs, which extended into the focal adhesions; notably, at focal adhesions EVL localization was predominantly basal to actin (Fig. 5e and Supplementary Fig. 6d).

Interestingly, while inhibition of contractility dissolved SCABs and increased protrusive activity in control cells, EVL-overexpressing cells exhibited a higher threshold of sensitivity

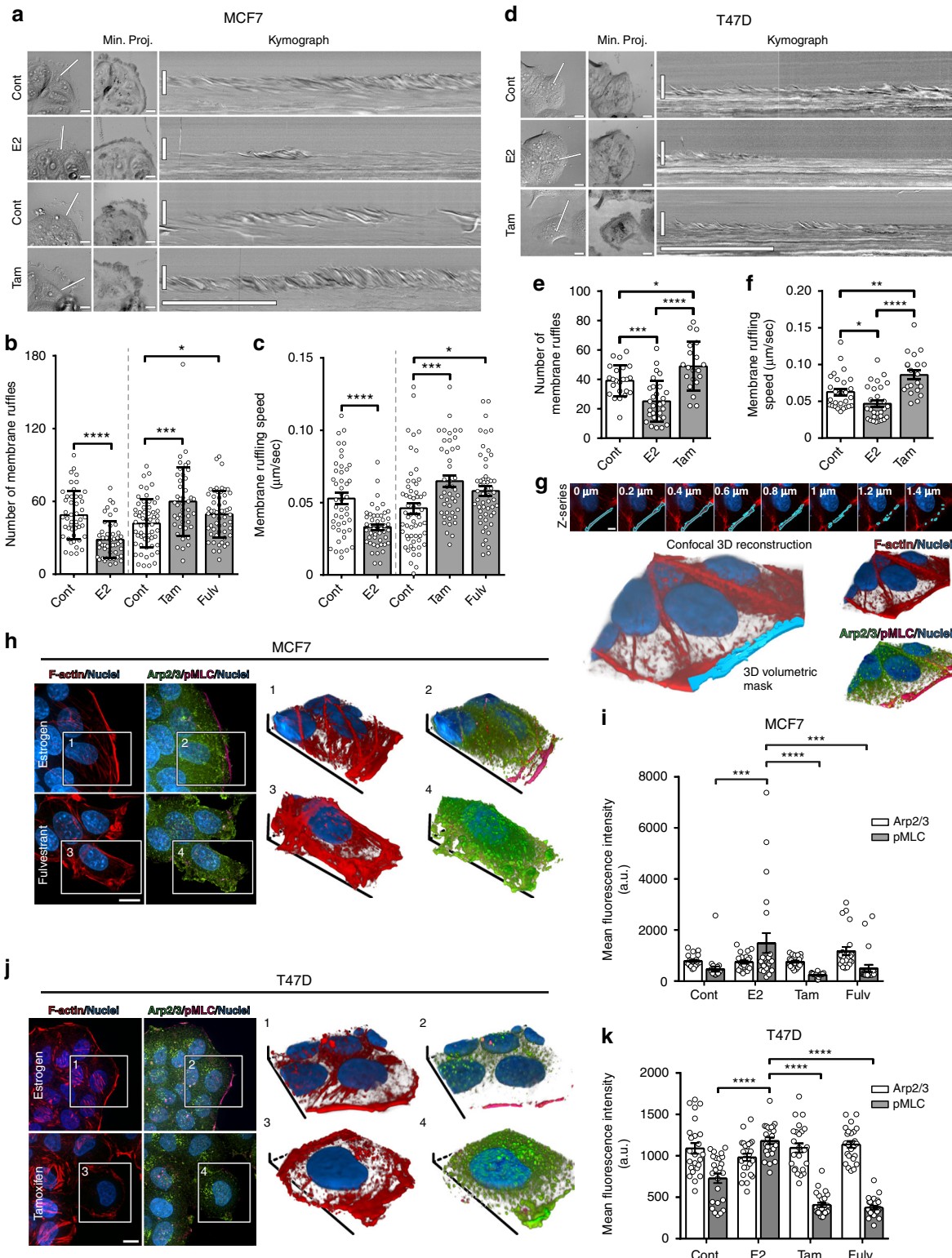

and their SCABs were more resistant to ROCK inhibition (Fig. 5f, Supplementary Fig. 5f, and Supplementary Movies 8, 9). This suggests that increased EVL expression not only leads to the enrichment of SCABs but also to the generation of more robust SCABs with higher efficacy in suppressing motility dynamics.

**EVL promotes ER-mediated suppression of invasion**. Importantly, EVL overexpression attenuated the increase in invasion induced by ER inhibition, whereas *EVL* KD reversed the suppressive effects of E2 on invasion (Fig. 6a, b). Moreover, EVL overexpression was sufficient to suppress invasion in ER− breast cancer SUM159 cells (Supplementary Fig. 7a, b). Together, these data suggest that EVL is necessary and sufficient to promote the E2-mediated suppression of invasion in vitro.

To extend these results in vivo, we examined the role of EVL in ER+ breast cancer progression (BCP) using an intraductal MCF7 xenograft model[39]. This model was used because it generates non-invasive ductal carcinoma in situ (DCIS) structures and maintains luminal differentiation, as opposed to fat-pad implantation, which induces mesenchymal differentiation[39]. Assessment of the intraductal tumors using a DCIS index (see Methods) showed that under E2 treatment, *EVL* KD constitutive (Fig. 6c and Supplementary Fig. 4a, b) and induced (Fig. 6d, e and Supplementary Fig. 7b) reduced the DCIS index and increased invasion compared to control. Consistent with the in vitro invasion data, these results show that EVL is necessary for mediating the suppression of invasion by ER in vivo.

**EVL level is inversely correlated with invasion in ER+ tumors**. Furthermore, analysis of immunofluorescence labeling of EVL and ER in luminal B tumor samples from TMA#2 resulted in the following findings: first, EVL immunolabeling demonstrated the presence of SCABs in tumors (Supplementary Fig. 7d); second, expression analysis revealed that the levels of EVL and ER are positively correlated, to the extent that, within tumors exhibiting heterogeneity in ER expression, EVL expression changes with that of ER at the single-cell level (Fig. 6f–h); and third, tumors expressing low EVL exhibited high LII (Fig. 6i, j) and more distant dissemination (EVL was ~30% less in tumors with LN >3, $p = 3 \times 10^{-5}$). Importantly, survival analysis showed that low EVL expression was associated with poor outcome in ER+ breast cancer in general, and in the luminal B subtype in particular (Fig. 6k and Supplementary Fig. 7e).

**EVL level decreased after anti-estrogenic hormone therapy**. We examined the effects of anti-estrogenic hormone therapy on EVL expression and invasion in ER+ breast cancer. We procured tumor samples from patients who were treated with neoadjuvant hormone therapy (Supplementary Data 8), and compared them to matching biopsy samples taken from the same tumors before therapy. We analyzed, at the single-cell level, invasion and EVL expression in a large population of tumor cells from three sets of before- and after-therapy tissue samples. Samples were collected from patients in stages 1, 2, and 3, thus covering a range of invasive behaviors from different stages of progression. Importantly, the single-cell analysis approach allowed us to capture intratumoral heterogeneity and examine changes in population dynamics in response to treatment. EVL labeling revealed that prolonged anti-estrogenic therapy led to a dramatic decrease in EVL expression, which was associated with a decrease in ER (Fig. 7a, b and Supplementary Fig. 8a, b). Analysis of local invasion (LII) showed that tumors became more invasive after treatment (Fig. 7c, d). Particularly, the portion of highly invasive cells (LII high) increased after treatment, especially in stage 1 and stage 2 tumors, which exhibited a lower number of invasive cells prior to treatment (Fig. 7c, d). These data suggest that anti-estrogenic hormone therapy, prescribed for these patients as neoadjuvant therapy to curb further growth of the tumors, potentially enhanced invasion with a marked decrease in EVL expression.

## Discussion
Despite promoting growth of ER+ tumors, estrogen suppresses their invasion. We determined that this significant, but poorly recognized protective role of estrogen is mediated by a distinct ER-dependent actin cytoskeletal architecture, reminiscent of actin organization in non-invasive epithelial cells[23]. As key features of these actin architectures, SCABs, which we denoted as SCABs, play a significant role in suppressing motility dynamics of the leading edge. Among a comprehensive list of actin cytoskeletal regulators, we found that *EVL* is the most differentially expressed in ER+ tumors, is a direct transcriptional target of ER, and is essential for ER-mediated actin cytoskeletal remodeling and suppression of invasion in vitro and in vivo. This study establishes a novel mechanism by which ER restricts dissemination of cancer cells and, potentially, inhibits their metastatic capacity (Fig. 8).

ER enriches for SCABs in ER+ breast cancer cells by promoting EVL expression, potentially as part of the luminal tran-

**Fig. 2** ER-mediated actin cytoskeletal remodeling induces suppressive cortical actin bundles (SCABs). **a** Leading edge kymography in representative time-lapse movies (Supplementary Movie 1) of control and E2-treated MCF7 cells (cultured under hormone starvation conditions), and control and ER-inhibited cells (cultured in regular media). Left panels indicate position at which kymographs were registered (line), and middle panels show minimum intensity projections from entire time series (Min. Proj.); scale bar is 10 μm. Right panels show corresponding kymographs; vertical scale bar is 10 μm; horizontal scale bar is 5 min. **b** Membrane ruffle quantification; mean ± s.d; *$p = 0.03$, ***$p = 0.0005$, ****$p < 0.0001$ (Welch's $t$ test). **c** Ruffling speed quantification; mean ± s.e.m; *$p = 0.01$, ***$p = 0.0007$, ****$p < 0.0001$ (Welch's $t$ test). Data from three independent experiments ($n \geq 45$ cells per group). **d** Leading edge kymography in representative time-lapse movies (Supplementary Movie 2) of T47D cells. **e** Membrane ruffle quantification; mean ± s.d; *$p = 0.03$, ***$p = 0.0002$, ****$p < 0.0001$ (Welch's $t$ test). **f** Ruffling speed quantification; mean ± s.e.m; *$p = 0.01$, **$p = 0.004$, ****$p < 0.0001$ (Welch's $t$ test). Data from two independent experiments ($n \geq 30$ cells per group). **g** Volumetric analysis of leading edge actin. Illustrative confocal z-series and corresponding 3D binary mask, demarcated as the area within 3 μm of cell edge and used to quantify leading edge fluorescence intensity. **h** Maximum intensity projections of z-series of E2-treated or fulv-treated MCF7 cells (Supplementary Movie 4), and 3D reconstructions of boxed areas; F-actin (red), Arp2/3 (green), and pMLC (magenta). Scale bar is 10 μm. **i** Quantification of leading edge Arp2/3 and pMLC in MCF7 cells. Data from three independent experiments ($n = 21$ cells per group; mean ± s.e.m). By two-way ANOVA, interaction analysis shows significant differences at $p < 0.0001$ (df = 3); pMLC values are significantly different in E2-treated group compared to other groups ($p < 0.001$). Arp2/3 values are not significantly different between groups. **j** Maximum intensity projections of z-series of E2-treated or tam-treated T47D cells and 3D reconstructions of boxed areas. F-actin is red, Arp2/3 in green, and pMLC in magenta. Scale bar is 10 μm. **k** Quantification of leading edge Arp2/3 and pMLC in T47D cells. Data from three independent experiments ($n = 21$ cells per group; mean ± s.e.m). By two-way ANOVA, interaction analysis shows significant differences at $p < 0.0001$ (df = 3); pMLC values are significantly different in E2-treated group compared to other groups ($p < 0.001$). Arp2/3 values are not significantly different between groups

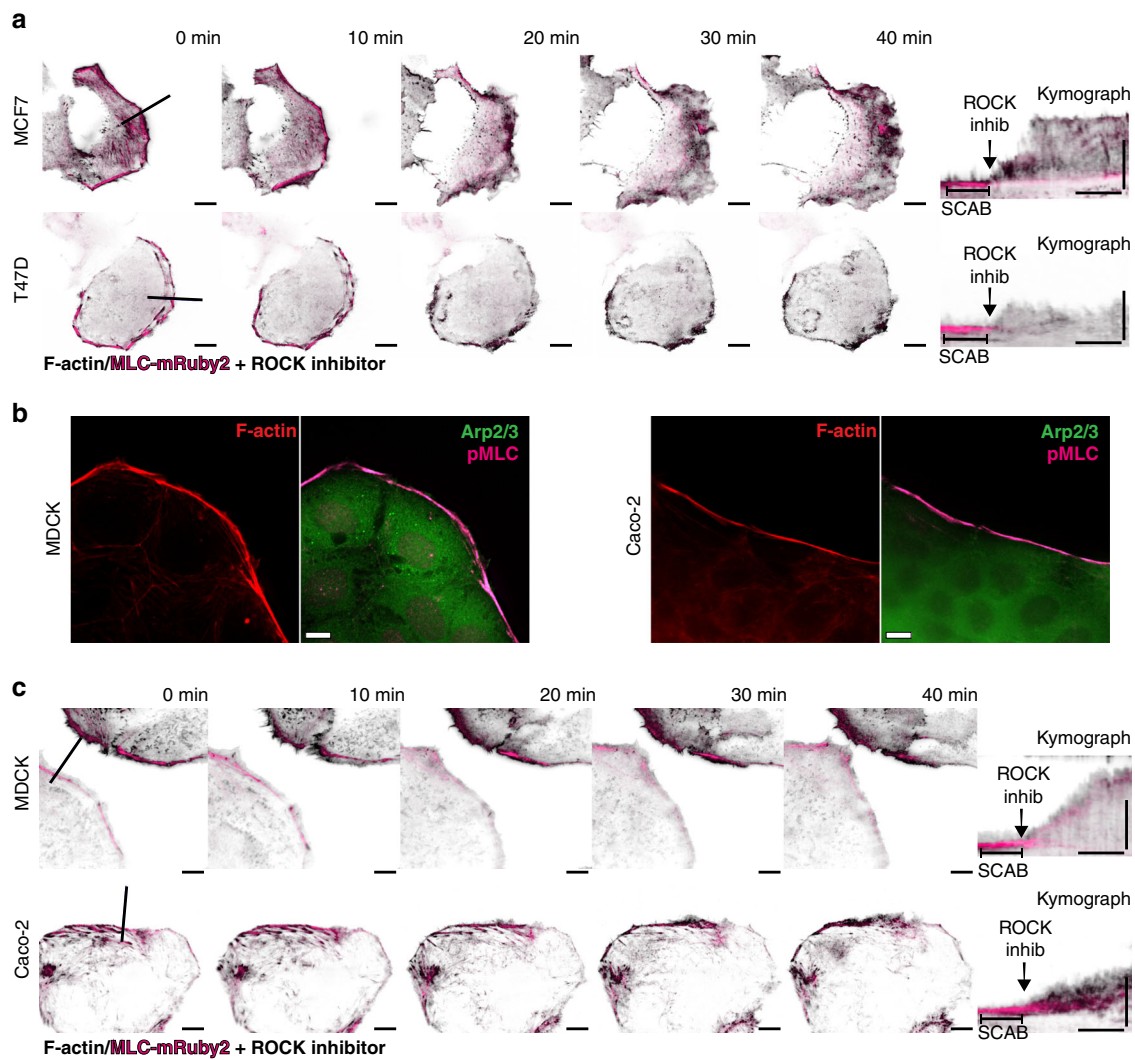

**Fig. 3** SCABs suppress leading edge motility in a myosin contractility-dependent manner. **a** Leading edge kymography in MCF7 cells (top row) and T47D cells (bottom row) expressing iRFP-Lifeact (black) and MLC-mRuby2 (magenta) before and after dissolving SCABs using 25 μM ROCK inhibitor. Left panels are images from TIRF microscopy time-lapse series before and after the addition of ROCK inhibitor (Supplementary Movie 5). Scale bar is 10 μm. Line shows the leading edge position at which kymographs were registered. Right panel shows MLC kymograph (denoting SCABs) superimposed over Lifeact kymograph (demarcating the leading edge). Vertical scale bar is 10 μm and horizontal scale bar is 10 min. **b** Immunofluorescence of SCAB in MDCK epithelial cells and Caco-2 human intestinal epithelial cells. Representative images for F-actin, pMLC, and Arp2/3. Scale bar is 10 μm. **c** Leading edge kymography in MDCK cells (top row) and Caco-2 (bottom row) cells expressing iRFP-Lifeact (black) and MLC-mRuby2 (magenta) before and after dissolving SCABs using 25 μM ROCK inhibitor. Scale bar is 10 μm. Line shows the leading edge location at which the kymograph was registered. Right panel shows MLC kymograph (denoting SCABs) superimposed over Lifeact kymograph (demarcating the leading edge). Vertical scale bar is 10 μm and horizontal scale bar is 10 min

scriptional program induced by ER in these cells. Indeed, EVL has been recently reported, by Tavares et al.[40], to be high in luminal breast tumors. Another actin cytoskeletal regulator, *VCL*, has been described as a transcriptional target of ER[20]. However, *VCL* was not up-regulated in any of the clinical datasets that we have analyzed, and its expression was not correlated with ER in ER+ tumors (Supplementary Data 3–6). Therefore, VCL is not expected to have a global role in keeping invasion in check in ER + tumors; *VCL* could potentially be down-regulated in only a subset of these tumors resulting in a specific mode of invasion, such as amoeboid invasion.

Interestingly, Tavares et al.[40] proposed that high levels of EVL are required for growth of pre-invasive tumors, but that *EVL* needs to be down-regulated in order to transition to an invasive state. These results are in line with our model and suggest that dissemination of ER+ breast tumors is potentially suppressed by high levels of EVL during tumor growth. Therefore, in ER+ breast cancer, the two central components of metastasis, tumor growth and dissemination, are potentially regulated by antagonistic mechanisms, suggesting that the dissemination step is distinct from the growth phase; this concept is consistent with the significant recurrence rate (30%) but long remission time

(>5 years) of this breast cancer subtype[41]. Our study highlights an emerging paradigm that deviates from the simplistic view that metastasis is mediated by the deregulation of a primary signal transduction pathway globally associated with malignant transformation; rather, for metastasis to happen, cancer cells need

reprogramming to acquire invasive properties and overcome antagonistic regulatory mechanisms promoting growth.

Importantly, most ER+ breast cancer patients receive extended adjuvant anti-estrogenic therapies. Despite the overwhelming evidence demonstrating the efficacy of these therapies in reducing

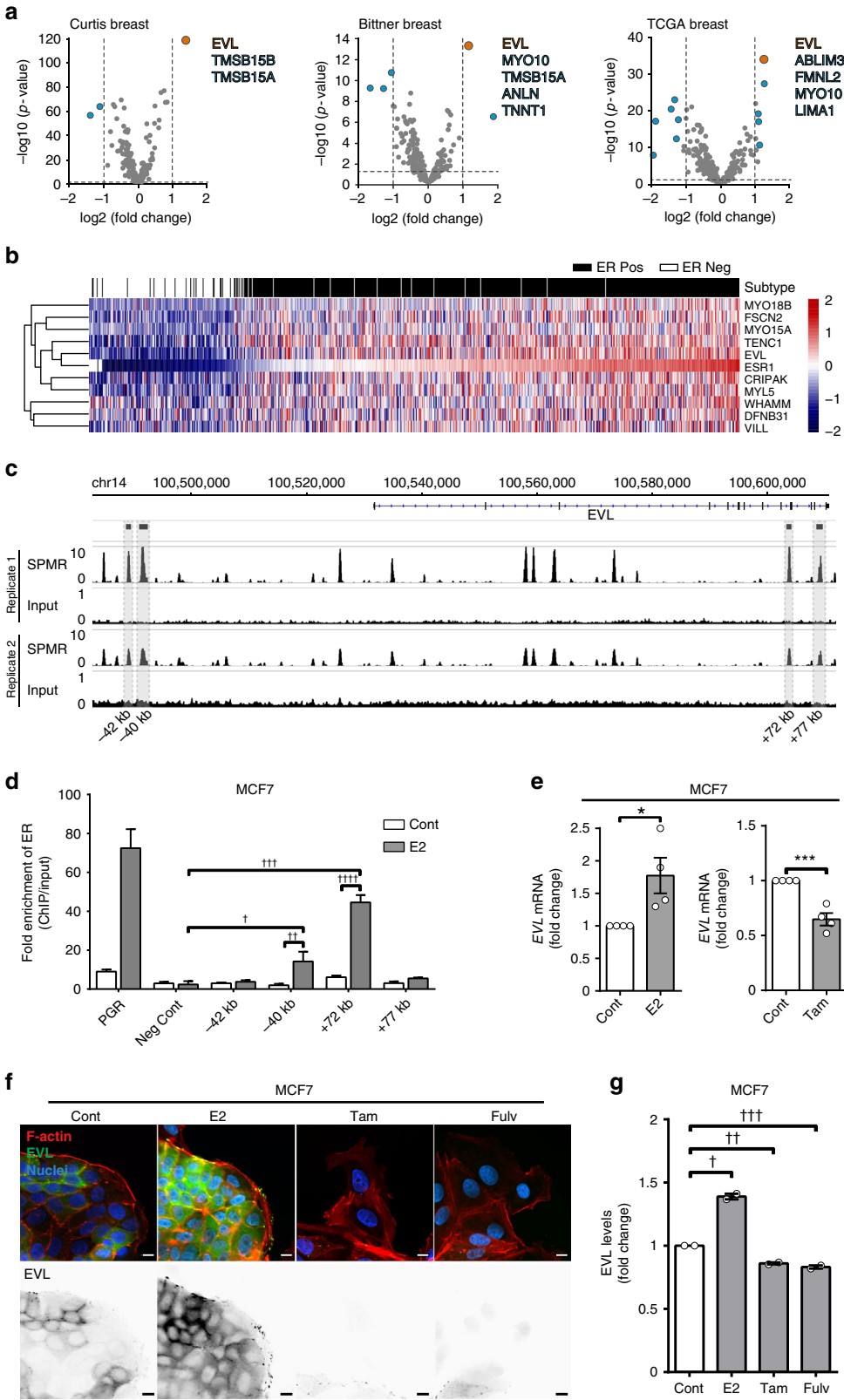

tumor recurrence, about one-third of treated patients are afflicted with recurrent, highly invasive tumors[41]. Our study raises the question of whether inhibiting ER could potentially down-regulate EVL and aggravate tumor invasion and dissemination of these recurrent tumors. Finally, the clinical significance of this work is in advancing our understanding of the protective actions of estrogen against ER+ BCP.

## Methods

**Antibodies and reagents**. The following antibodies were used: human anti-(Thr18/Ser19) pMLC 2 no. 3674 (Cell Signaling), hHuman anti-Arp2/3 clone no.13C9 (Millipore), human anti-pan cytokeratin no. AE1/AE3 (Dako), human anti-ER no. H4624 (Invitrogen), human anti-EVL no. HPA018849 (Sigma Prestige), human anti-EVL (a gift from Frank Gertler), and human anti-Actin (Abcam, ab3280). The following reagents were used: Alexa Fluor 568-Phalloidin (Thermo Fisher), Alexa Fluor 647-Phalloidin (Thermo Fisher), Hoechst 33342 (Thermo Scientific), β-estradiol (Sigma-Aldrich), tamoxifen (tam) (Sigma-Aldrich), fulvestrant (fulv) (Sigma-Aldrich), ROCK inhibitor (γ-27632; Tocris), insulin growth factor 1 human recombinant (Sigma-Aldrich), puromycin (Thermo Scientific), G418 (Life Technologies), polybrene (EMD Millipore), and Fugene HD (Life Technologies).

**Cell culture**. MCF7 and HEK293T cells were cultured in high glucose Dulbecco's modified Eagle's medium (DMEM) base media with sodium pyruvate and L-glutamine (Corning) supplemented with additional 100 mM L-glutamine, 10% fetal bovine serum (FBS; Gibco), and antibiotics (100 U/mL penicillin + 100 μg/mL streptomycin from Life Technologies, Inc.). T47D were cultured in RPMI1640 base media with L-glutamine (Corning) supplemented with additional 100 mM L-glutamine, 5 μg/mL insulin (Roche), 10% FBS, and antibiotics. MCF10A were cultured in DMEM/F12 50/50 with L-glutamine (Corning), 5% horse Serum, 20 ng/mL epidermal growth factor (Peprotech), 0.5 mg/mL hydrocortisone (Sigma), 100 ng/mL cholera toxin (Sigma), 10 μg/mL insulin, and antibiotics. Caco-2 were cultured in minimum essential media (MEM) base media with L-glutamine (Corning), 20% FBS, and antibiotics. MDCK cells were cultured in high glucose DMEM base media with sodium pyruvate and L-glutamine (Corning), 10% FBS, and antibiotics.

**Plasmids**. The following plasmids were used for RNAi: pLKO.1-TRC cloning vector was a gift from David Root (Addgene, plasmid #10878), pLKO EVL shRNA (GE Dharmacon, TRCN0000063869, antisense TACTAGGATCTTCCATTTGGC), TRIPZ (GE Dharmacon, RHS4750), and TRIPZ EVL shRNA (GE Dharmacon, PN V3THS_300209, antisense TGGCTTTCATCTTCCTTCT), previously validated[26]. MSCV-eGFP-EVL was a gift from Frank Gertler, MIT, used for retroviral expression in kymography experiments. Lentiviral construct pLenti CMV-MRLC1-mRuby2-IRES-PuroR was generated by PCR amplification of complementary DNA (cDNA) for Homo sapiens MRLC1 from eGFP-MRLC (a gift from Tom Egelhoff, Addgene, #35680) and mRuby2-N1 (a gift from Michael Davidson, Addgene, #54614) to create C-terminal-tagged MRLC. Fusion construct was subcloned into the expression vector pCIG3 (pCMV-IRES-GFP, a gift from Felicia Goodrum, Addgene, plasmid #78264), modified to replace the GFP cassette with puromycin resistance gene. pCMV-iRFP670-EVL-IRES-BlastR was generated by PCR amplification of cDNA for Mus musculus EVL cDNA from MSCV-eGFP-EVL and piRFP670-N1 (a gift from Vladislav Verkhusha, Addgene, #45457) to create N-terminally tagged EVL. Fusion construct was subcloned into pCIG3 modified to replace GFP with blasticidin resistance. Lentiviral expression constructs pLenti EF1a-eGFP-EVL, pLenti EF1a-eGFP, pLenti Lifeact-iRFP670-BlastR, and pLenti

Lifeact-eGFP-BlastR were generated by Gateway technology (Thermo). pLenti EF1a-eGFP-EVL and pLenti EF1a-eGFP were generated by subcloning M. musculus EVL cDNA from MSCV-eGFP-EVL into eGFP-C2 (Clontech). eGFP and eGFP-EVL were subcloned into Gateway entry vector pMuLE ENTR MCS L5-L2 (a gift from Ian Frew, Addgene, plasmid #62085) modified to contain the hEF1a promoter, and recombined with pMuLE ENTR MCS L1-R5 (a gift from Ian Frew, Addgene, plasmid #62084) and pLenti Dest PuroR R1-R2 using LR Clonase II (Thermo) to generate the final lentiviral expression vector. pLenti Lifeact-eGFP-BlastR and pLenti Lifeact-iRFP670-BlastR were generated by subcloning Lifeact-mEGFP (a gift from Michael Davidson, Addgene, #54610), or Lifeact-iRFP670 into pMuLE ENTR MCS L5-L2, and recombined with pMuLE ENTR MCS L5-L2 and pLenti Dest BlastR. The following plasmids from this paper have been deposited to Addgene: pLenti CMV-MRLC1-mRuby2-IRES-PuroR (#103031), Lifeact-iRFP670 (#103032), pLenti Lifeact-iRFP670-BlastR (#84385), pLenti Lifeact-eGFP-BlastR (#84383), pENTR CMVie-Lifeact-iRFP670 L1-R5 (#84390), pENTR CMVie-Lifeact-EGFP L1-R5 (#84391), pLenti Dest BlastR R1-R2 (#84574), and pLenti Dest PuroR R1-R2 (#84575).

**Virus production**. HEK293T cells were transfected at 60% confluence using Fugene HD (Promega) in OptiMEM (Corning) with transfer plasmid and second-generation lentiviral packaging system (psPAX2 and pMD2.G, Addgene #12260 and #12259, gifts from Didier Trono) or pCL-Ampho (Novus) for lentiviral or retroviral production, respectively. Virus was collected 48–72 h post transfection, clarified by 0.45-μm filters. Recipient cells were infected at 50% confluence with virus at a 1:1 dilution with their culturing media containing polybrene (10 μg/mL). Puromycin selection was started 36 h post infection.

**Analysis of LN dissemination**. LN dissemination was determined in a meta-analysis of epidemiological studies[6–15] that investigated the effects of HRT on the clinical and prognostic characteristics of tumors in cohorts of breast cancer patients. Specifically, only studies that reported LN status in HRT current-users and never-users, with a minimum of 10 patients per treatment group, were included. Studies that investigated the effect of HRT on breast cancer incidence in healthy women were not considered. This focused approach enriched for breast cancer patients on HRT, providing sufficient data for a robust quantification of LN dissemination under estrogen treatment. For patient inclusion criteria, the analysis included only postmenopausal breast cancer patients (older than 50 years of age and had at least 6 months of amenorrhea or had double oophorectomy), who were on HRT for at least 6 months before diagnosis (denoted as current-users) or who never used HRT (denoted as never-users); HRT included treatment with estrogen alone or estrogen plus progestin. Patients taking estriol, a partial ER agonist available as a dietary supplement, were excluded from our analysis. This meta-analysis consisted of two parts: calculating OR for having more than three positive LN (LN >3), using six studies[6,8,9,11,12,15]; and calculating OR for having any number of positive LN (LN+), using nine studies[7–15]. For both analyses, Fisher's exact test was conducted to determine significance of each study. The OR was determined for each study and the exact 95% confidence interval was calculated using SATA 14.0.

**Tissue immunolabeling**. Slides were baked for 1 h at 60 °C, deparaffinized in xylene, and rehydrated in increasingly diluted ethanol solutions. Heat-induced epitope retrieval was performed (40 min boiling + 20 min cooling) in Tris-EDTA pH 9 buffer for anti-ER (Invitrogen, 1:50 dilution) and anti-EVL (Sigma Prestige, 1:50 dilution) antibodies and sodium citrate pH 6 buffer for anti-pan cytokeratin AE1/AE3 (Dako, 1:100 dilution). Following standard immunofluorescence staining protocol, samples were mounted in ProLong Diamond Antifade (Thermo Fisher) and allowed to cure for at least 24 h before imaging.

---

**Fig. 4** *EVL* is a transcriptional target of ER. **a** Differential gene expression analysis of actin cytoskeletal regulators in ER+ vs. ER− breast tumors. Volcano plot of significance ($-\log_{10}$ of $p$ value) vs. $\log_2$ fold change. Horizontal dashed line represents threshold for significance at $p < 0.05$. Vertical dashed lines represent threshold for positive or negative twofold change in gene expression. Insets are lists of genes that passed both thresholds in descending order of significance (lists were limited to five genes; full data analyses are presented in Supplementary Data 3). **b** Unsupervised clustering analysis of TCGA RNA-seq dataset. Heatmap, generated by row scaling, showing expression of genes in the *ESR1* cluster. Tumor samples were classified into ER+ (black) and ER− (white) based on ER status (paired $t$ test used to determine significance). **c** Analysis of ER binding by chromatin immunoprecipitation (ChIP). Genomic region surrounding *EVL* showing ER-ChIP sequencing profile as SPMR (signal per million reads) trace of two independent ER-ChIP sequencing samples, generated by MACS2.0 and visualized by IGV (Integrative Genomics Viewer). Corresponding input profiles are shown at ×10 scale to clearly show the input DNA openness bias. **d** Fold enrichment of ER binding at four peaks found by ChIP-seq, in addition to positive (PGR) and negative controls. Fold enrichment is calculated as [DNA]-normalized ChIP/Input of $2^{\Delta\Delta Ct}$. Values are means of data from four replicates; mean ± s.e.m.; $p$ values were generated by paired $t$ test comparing E2 treated to vehicle treated and E2 treated to negative control; †$p = 0.009$, ††$p = 0.007$, †††$p = 0.001$, ††††$p = 0.001$. **e** Analysis of the Regulation of EVL expression by ER. qPCR of *EVL* mRNA in MCF7 cells treated with corresponding drugs for 24 h, normalized to GAPDH. Fold change shown between treatment groups. Values are means of data from four independent experiments; mean ± s.e.m.; *$p = 0.03$, ***$p = 0.0008$ (unpaired $t$ test). **f** Immunolabeling of MCF7 cells after corresponding drug treatment for 72 h. Bottom row shows EVL labeling. Scale bar is 10 μm. **g** Quantification of EVL levels in MCF7 cells after treatment with corresponding drugs for 72 h. Values are means of data pooled from two independent experiments; mean ± s.e.m. †$p = 0.03$, ††$p = 0.03$, †††$p = 0.04$ (Welch's $t$ test)

Imaging: Tumors were imaged using a ×20 Plan Apo 0.75NA Nikon objective on a Nikon Ti-E inverted microscope equipped with a Hamamatsu ORCA-Flash 4.0 V2 cMOS camera, motorized stage, and maintained in focus using Nikon Perfect Focus System. Large composite images of whole tumor sections were generated by digital stitching using Nikon Elements. Tumor sections that were damaged or that exhibited high degree of inflammatory cell infiltration were excluded from the analysis.

**LII analysis.** Breast cancer TMAs: Two independent breast cancer TMA sets were used in this analysis: (TMA#1) Cancer Diagnostics Program (CDP) BCP TMA from the National Cancer Institute (NCI), comprised of an array of ER+ and ER− tumors; and (TMA#2) luminal B breast cancer TMA from Cedars-Sinai Medical Center (Dr. Johnathon Kaye), comprised of an array of luminal B breast cancer tumors with varied levels of ER expression. LN status was indicated in the corresponding pathology report for each patient. Samples that were damaged or that exhibited high degree of inflammatory cell infiltration were excluded from the analysis.

Matched before and after endocrine therapy patient samples: Samples were obtained from the Arizona Cancer Center Biospecimen Repository. Samples were collected in compliance with the University of Arizona Institutional Review Board policies.

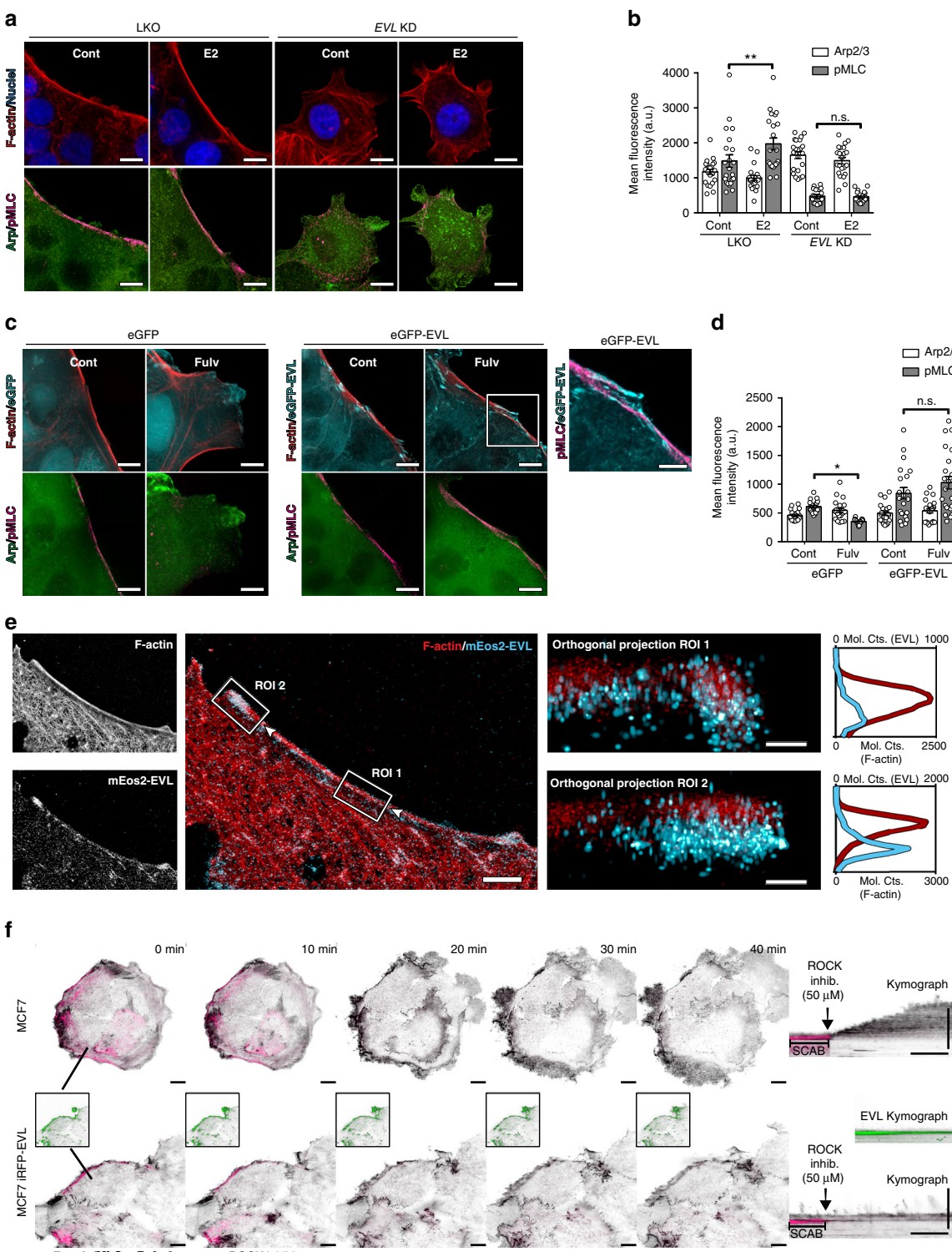

LII measurement: Cancer cell invasion was quantified in both TMAs (64 samples from the NCI CDP-BCP-TMA and 180 samples from the Cedars-Sinai LumB TMA) by employing the NND approach. Cancer cells were identified using a binary mask generated by thresholding the fluorescence signal of the cytokeratin stain. Using a built-in module in Nikon Elements, the distance between the nucleus of each cancer cell and the nucleus of the cell nearest to it was measured. Cells scattered at the periphery of the analyzed region were excluded from our analysis if their NND zone (circular zone with centroid as center and NND as radius) is not fully contained within the analyzed area. LII was calculated as the average NND of all cancer cells within the analyzed tumor area.

**Three-dimensional culture system**. A tissue culture model system was designed to quantitate 3D invasion of cells with low invasive capacity; the configuration of this system allowed reliable monitoring of invading cells over a relatively long period of time (≥2 days). The system was setup in 8-well glass-bottom chamber slides (Labtek), where cancer cells were embedded in a central 3D matrix and allowed to invade into an encapsulating outer 3D matrix. Prior to utilization, the bottom coverslips of the chamber slides were silanized using 2% aminopropyl-trimethoxysilane (Acros Organics) in isopropanol for 10 min at room temperature, followed by repeated water washes and drying at 37 °C. Chamber slides were then treated with 1% glutaraldehyde for 20 min at room temperature, followed by water washes and drying at 37 °C, and finally sterilization under ultraviolet light for 15 min. After corresponding pretreatments, $1 \times 10^5$ cells per well were resuspended in 10 μL of 1 mg/mL type-I collagen containing 1.25 mg/mL polyethylene glycol diglycidyl ether (Fisher). Cell suspension was added to the center of the chamber wells and allowed to polymerize in the tissue culture incubator. The outer matrix, prepared similarly to the collagen mixture described above with the addition of 20 nM human recombinant IGF1, was added to encapsulate the central matrix; after polymerization, the setup was overlaid with growth media containing the corresponding treatments. Cells were allowed to invade into the outer matrix for 48 h (with one media change at 24 h), and then they were fixed and stained with Hoechst 33342 (Thermo Scientific) for 24 h at 4 °C. Confocal z-series of Hoechst and differential interference contrast (DIC) were acquired at 1.5 μm z-steps on a Nikon Ti-E inverted microscope equipped with a ×20 Plan Apo 0.75NA objective, Photometrics CoolSNAP MYO CCD camera, motorized stage, and a Crest XLIGHT spinning-disk confocal system. At each z-step, large-image stitching was used to generate a composite image of the entire area including the central matrix and the invaded cells. A binary mask was generated from the maximum intensity projection image of the z-series. A region of interest (ROI) was created by excluding the central area (determined using the overlaid DIC image) and the number of invading cells in the ROI was quantitated using the Object Count Nikon Elements module.

**Kymography analysis**. DIC microscopy time-lapse imaging: Cells were plated at 30% confluence on glass-bottom 6-well plate (MatTek) and allowed to adhere for 24 h before experimentation. Cells were treated with E2 (10 nM), tam (1 μM), or fulv (100 nM) in complete growth media, which were replaced every 24 h. Cells were imaged 72 h after plating at 1 frame/s for 45 min using DIC microscopy. Images were acquired with a ×60 Plan Apo 1.40 NA Nikon objective on a Nikon Ti-E inverted microscope equipped with a Hamamatsu ORCA-Flash 4.0 V2 cMOS camera, environmental chamber, motorized stage, and maintained in focus using Nikon Perfect Focus System.

Kymography analysis of membrane ruffles: A kymograph was registered along the axis of ruffling at the site of highest membrane dynamics (as determined by minimum intensity projections). On the generated kymograph, membrane ruffles were identified as areas of high contrast. For each kymograph, two parameters of membrane ruffles were analyzed: frequency and speed. Frequency was determined by quantitating ruffle number (areas of high contrast) in each kymograph. Speed was determined as the distance traveled by a ruffle/time.

**Immunofluorescence staining**. Cells were fixed with 4% paraformaldehyde (PFA) (Electron Microscopy Sciences) for 10 min at 37 °C. PFA was quenched with 0.1 M glycine and cells were permeabilized with 0.1% Triton X-100, both for 10 min at room temperature. The actin cytoskeleton was stained with Alexa Fluor-conjugated phalloidin (Alexa Fluor 568-Phalloidin or Alexa Fluor 647-Phalloidin; Thermo Fisher) diluted to 2.5% in phosphate-buffered saline (PBS); nuclei were stained with Hoechst 33342 (Thermo Scientific); pMLC was immunolabeled using human anti-(Thr18/Ser19) pMLC 2 no. 3674 (Cell Signaling) at 1:100 dilution; Arp2/3 was immunolabeled using human anti-Arp2/3 clone no.13C9 (Millipore) at 1:200 dilution; EVL was immunolabeled using human anti-EVL (a gift from Frank Gertler) at 1:50 dilution. Slides were blocked in 1% FBS/1% bovine serum albumin (BSA) in PBS; primary and secondary antibodies were diluted in blocking buffer and incubated using standard protocols. Samples were mounted in ProLong Diamond Antifade (Thermo Fisher) and allowed to cure for at least 24 h before imaging.

**Volumetric analysis**. Cells were fixed with 4% PFA (Electron Microscopy Sciences) for 10 min at 37 °C. PFA was quenched with 0.1 M glycine and cells were permeabilized with 0.1% Triton X-100, both for 10 min at room temperature. The actin cytoskeleton was stained with Alexa Fluor-conjugated phalloidin (Alexa Fluor 568-Phalloidin or Alexa Fluor 647-Phalloidin; Thermo Fisher) diluted in PBS; nuclei were stained with Hoechst 33342 (Thermo Scientific). Slides were blocked in 1% FBS/1% BSA in PBS at room temperature for 1 h. pMLC and Arp2/3 were immunolabeled with primary and secondary antibodies diluted in blocking buffer and incubated using standard protocols. Confocal z-series were acquired at 0.2 μm z-steps on a Nikon Ti-E inverted microscope equipped with a ×100 Plan Apo TIRF 1.49NA Nikon objective, Photometrics CoolSNAP MYO CCD camera, motorized stage, a Crest XLIGHT spinning-disk confocal system, and a Lumencor SPECTRA X light source. Mean fluorescence intensity of pMLC and Arp2/3 at the leading edge was determined within a 3D volumetric binary mask, generated using the fluorescent signal of the actin stain. Nikon Elements 3D Object Module was used to construct the 3D binary mask from a series of 2D masks generated at each z-step in the actin channel. Importantly, to limit the analysis to the leading edge actin, the masks were generated within an ROI that was restricted to a 3-μm-wide band juxtaposed to the membrane.

**TIRFM time-lapse imaging**. Cells expressing Lifeact-iRFP670 and MLC-mRuby2 were used in this experiment to analyze actin dynamics during protrusion at the leading edge. Cells were plated as described above and ROCK inhibitor was added at indicated concentrations to dissolve SCABs. Cells were imaged using TIRF illumination with a ×100 Plan Apo TIRF 1.49NA objective on a Nikon Ti-E inverted microscope equipped with a Hamamatsu ORCA-Flash 4.0 V2 cMOS camera, Photometrics Evolve 512 EMCCD camera. Images were acquired on a 20-s interval for indicated duration. For EVL overexpression time-lapse experiments, cells expressing Lifeact-mEGFP and MLC-mRuby2, with or without iRFP670-EVL, were co-cultured and imaged as described above.

Kymography analysis of actin dynamics: Similar approach as above was used to register the kymographs, with the exception that maximum intensity projection of the fluorescent images from the entire time series was used to guide the positioning of the kymograph line.

**Differential gene expression analysis of actin cytoskeletal regulators**. GO Consortium search engine was used, with the search term "cytoskeleton," to

**Fig. 5** EVL promotes ER-mediated actin remodeling. **a** Maximum intensity projections of z-series of control and E2-treated LKO and *EVL* KD MCF7 cells. Scale bar is 10 μm. **b** Quantification of leading edge Arp2/3 and pMLC. Data from three independent experiments ($n = 21$ cells per group; mean ± s.e.m.). Three-way ANOVA shows significant differences between LKO and *EVL* KD groups at $p = 0.08$ (df = 1). Two-way ANOVA shows significant difference in pMLC levels at $p < 0.05$ between control and E2-treated cells in the LKO group, but not in the *EVL* KD group. No significant differences in Arp2/3 levels were observed at $p < 0.05$. **p < 0.01. **c** Maximum intensity projections of z-series of control and fulv-treated eGFP and eGFP-EVL MCF7 cells. Scale bar is 10 μm. Scale bar is 5 μm for inset showing eGFP-EVL and pMLC. Larger fields of view from the same group are given in Supplementary Fig. 5b. **d** Quantification of leading edge Arp2/3 and pMLC. Data from three independent experiments ($n = 21$ cells per group; mean ± s.e.m.). Three-way ANOVA shows significant differences between eGFP and eGFP-EVL groups at $p < 0.0001$ (df = 1). Two-way ANOVA shows significant difference in pMLC levels at $p < 0.05$ between control and fulv-treated cells in the eGFP group, but not in the eGFP-EVL group. No significant differences in Arp2/3 levels were observed at $p < 0.05$. *p < 0.05. **e** Analysis of EVL localization at SCABs by iPALM in MCF7 cells expressing mEos2-EVL. Left panels show single and merged images of F-actin staining and mEos2-EVL; boxes indicate ROIs. Arrows indicate en face orientation of orthogonal projections. Orthogonal projections of ROIs are shown with corresponding histogram highlighting z-localization of F-actin (red) and mEos2-EVL (cyan); horizontal axes represent molecular counts (mol. cts.) for mEos2-EVL (top) and F-actin (bottom). Merge image scale bar is 5 μm. Orthogonal projection scale bar is 150 nm. **f** Leading edge kymography in control and iRFP670-EVL (green) expressing MCF7 cells (top and bottom rows, respectively), with eGFP-Lifeact (black) and MLC-mRuby2 (magenta), before and after ROCK inhibitor (50 μM) treatment (Supplementary Movie 8). Left panels are images from time-lapse series before and after treatment. Scale bar is 10 μm. Line shows leading edge location at which kymographs were registered. Inset shows EVL channel separately. Right panel shows kymograph. Vertical scale bar is 10 μm. Horizontal scale bar is 10 min. n.s. nonsignificant

generate a list of genes associated with the cytoskeleton. The initial list of 1000+ genes retuned in the GO search (www.geneotology.org) was further refined by selecting only cytoskeletal regulators that directly interact with actin, as determined by hand for each gene from an extensive literature review; this process generated a shorter list of 285 actin cytoskeletal regulators (Supplementary Data 1). Using public datasets available on Oncomine (www.oncomine.org), differential expression in ER+ breast cancer of the genes in our curated gene set was analyzed. The following criteria for inclusion of studies were applied: (1) studies must have included clinical samples from patients (studies with only cell line data were excluded); (2) studies must have included both ER+ and ER− tumors, with a minimum of ten samples per group; and (3) microarray chips used in the studies must have had probes representing a minimum of 95% of the genes on our curated gene set. Based on these criteria, we selected 12 out of the 56 studies available on Oncomine. Data were presented in volcano plots to visualize both fold change and significance, concomitantly. Our thresholds for identifying a gene as a positive "hit" were twofold change in expression (in either positive or negative direction) and $p \leq 0.05$.

**RNA-seq analysis**. We analyzed mRNA expression in breast cancer in TCGA breast cancer RNA-seq dataset. RNASeq Version 2 level 3 data were downloaded from TCGA breast cancer; 1062 breast tumors and 113 normal tissue samples were included in the analysis. Tumor samples were classified as ER+ or ER− based on the ER status indicated in the accompanying clinical report. Fold change was computed manually using R as well as via bioconductor package, DESeq2. Raw counts from level 3 data, which includes non-integer numbers, were rounded off for analysis in DESeq2, which requires integer-based read counts. $P$ value was calculated using paired $t$ test for manual calculation and Benjamini–Hochberg method was used for multiple test corrections. The bioconductor package, pheatmap3, was used to plot heat maps using row scaling on gene counts and clustering was enabled for gene clustering

**ChIP assay**. Experiments were performed as described previously[42]. Briefly, prior to the assay, cells were hormone-starved for 3 days in phenol-free DMEM base media supplemented with 10% charcoal-stripped FBS. Starved cells were treated for

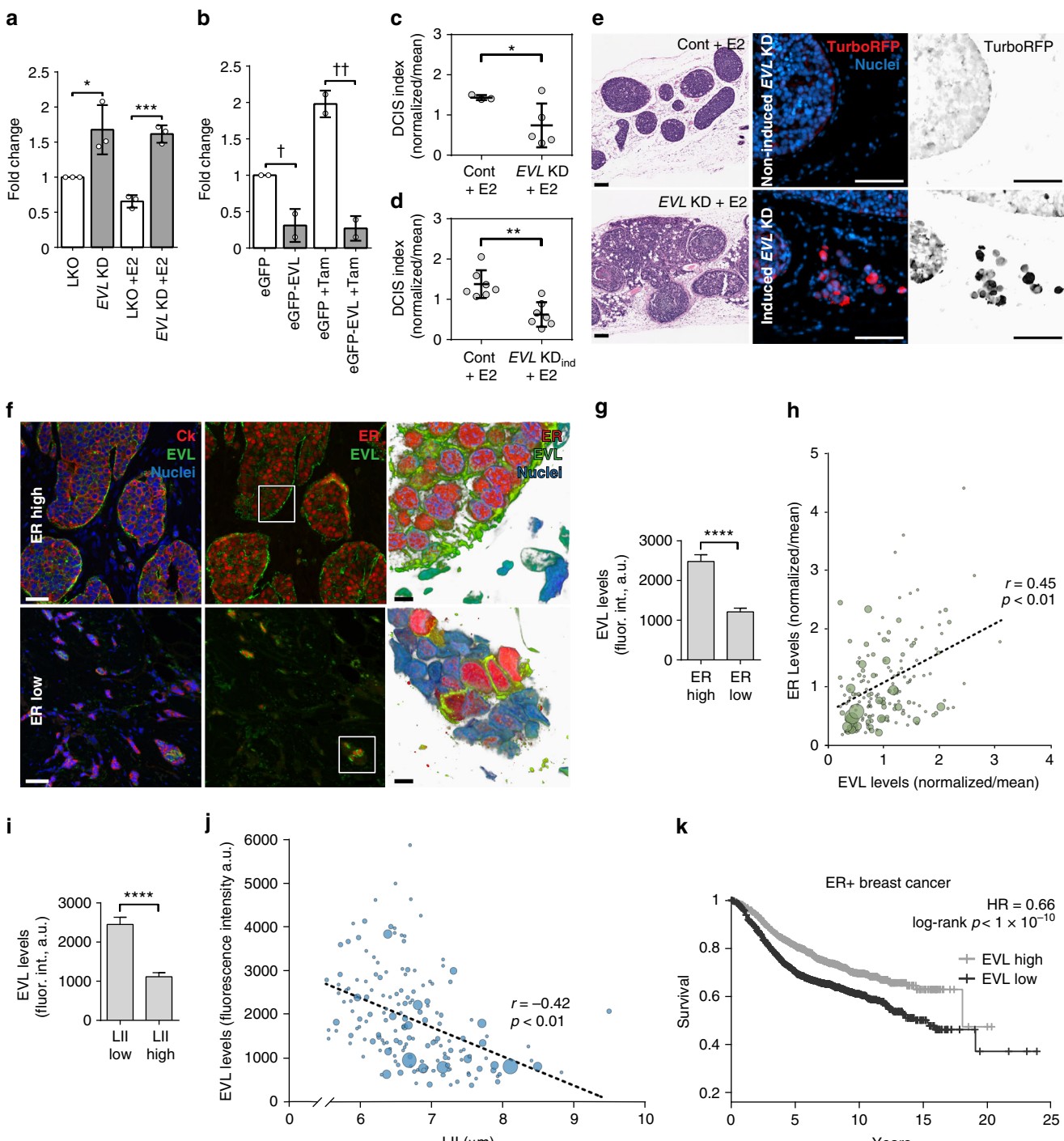

45 min with E2 (10 nM). For ChIP, chromatin was cross-linked with 1% PFA (Electron Microscopy Sciences) for 8 min at room temperature. PFA was then quenched using 125 mM glycine and 5 mg/mL BSA. Cells were scraped with PBS and cellular pellets were isolated by centrifugation at 2000 rpm for 5 min at 4 °C. Pellets were resuspended in lysis buffer (1% sodium dodecyl sulfate (SDS), 10 mM EDTA and 50 mM Tris, pH 8.0) and sonicated using an E210 instrument (Covaris) to an average size of 350 bp according to the manufacturer's protocol. After clearing by centrifugation at 15,000 rpm for 15 min, 5 μl of sample was collected as input before immunoprecipitation, and the rest was diluted fivefold in dilution buffer (20 mM Tris, pH 8.0, 150 mM NaCl, 2 mM EDTA, and 1% Triton X-100). Chromatin complexes were immunoprecipitated by an overnight incubation at 4 °C with 2 μg anti-ER antibodies (Santa Cruz, HC-20 and Thermo Fisher, AB-10), followed by a 45 min incubation with 40 μl 1:1 mix of protein A and G Dynabeads (Invitrogen). Beads were then washed four times with RIPA lysis buffer (50 mM HEPES, 1 mM EDTA, 0.7% deoxycholate, 1% NP-40, and 0.5 M LiCl) and twice with Tris-EDTA (TE) buffer. The protein–DNA complexes were eluted in 1% SDS, 0.1 M sodium bicarbonate buffer heated at 65 °C for 6 h to reverse the PFA cross-linking. DNA was isolated using PCR Cleanup Kit (Qiagen) and quantitated using Quant-IT as per the manufacturer's instructions (Thermo Scientific). DNA fragments were analyzed by qPCR using SYBR Green Master Mix (Kapa Biosciences) on an ABI 7300 instrument (Applied Biosystems). For ChIP-seq, purified DNA was then end repaired and ligated to adapter oligos (Illumina) and amplified. Sequencing was performed at the Center for Cancer Computational Biology at Dana Farber Cancer Institute using an Illumina HiSeq2000 instrument. Reads were mapped to the human genome using STAR and peaks indicating ER binding were called comparing ChIP and corresponding Chromatin Input libraries using MACS2.1 and a threshold of $p < 10^{-5}$. ChIP-seq SPMR (signal per million reads) traces were generated using MACS2.1 followed by conversion to BigWig format using the bedGraphToBigWig tool. High confidence binding cutoff was arbitrarily chosen at $q < 10^{-200}$ and presence in both ChIP-seq replicates corresponding to approximately 10% of the entire dataset.

**Quantitative real-time PCR**. Cells were treated for 24 h with E2 (10 nM), tam (1 μm), or fulv (100 nM); E2-treated cells were hormone-starved for 24 h as described above. Total RNA was isolated using Isolate II RNA Kit (Bioline) and cDNA was then synthesized from 1 μg of RNA using XLA Script cDNA Kit (Quanta BioSciences). SYBR Green PCR Mix (Apex) was used for Real Time qPCR on the ABI Fast 7500 system. Samples were run in triplicates in each experiment and relative mRNA levels were normalized to *GAPDH* (glyceraldehyde 3-phosphate dehydrogenase). qPCR primer sequences are found in Supplementary Table 9.

**Quantification of EVL protein by immunofluorescence**. Cells were fixed and stained as described above and imaged using a ×20 Plan Apo 0.75NA Nikon objective on a Nikon Ti-E inverted microscope equipped with a Hamamatsu ORCA-Flash 4.0 V2 cMOS camera, motorized stage and maintained in focus using Nikon Perfect Focus System. For each drug treatment, two large composite images of 8 × 8 fields of view were generated by digital stitching using Nikon Elements. To quantify EVL levels, a binary mask was generated against F-actin staining using Nikon Threshold Module and EVL signal was quantified within this mask.

**DCIS mouse model**. A previously described protocol of intraductal injection was followed with modifications[43,44]. In brief, 8-week-old female nonobese diabetic/severe combined immunodeficiency mice were ovariectomized and implanted with a sub-dermal 90-day release 0.72 mg E2 pellets (Innovative Research of America). Two days after implantation, $1 \times 10^4$ cells suspended in 3 μL growth media were injected into the primary mammary duct through a cleaved nipple, using a

Hamilton syringe. For the first experiment, we used MCF7 cells expressing shRNA targeting *EVL* and control cells expressing the LKO vector control; for the second experiment, we used cells expressing an inducible form of shRNA targeting *EVL* and TRIPZ vector control. After 6–8 weeks, tumors were harvested, fixed in 4% PFA (Electron Microscopy Sciences), and processed for tissue staining. For experiments with cells expressing an inducible TRIPZ constructs, doxycycline was administered in the drinking water at a concentration of 2 mg/mL for 2 weeks before tumor harvesting. All experiments were performed according to the guidelines of the IACUC committee of the University of Arizona.

**Analysis of DCIS index**. Quantification of DCIS index: Large composite images of the entire mid-sections from each tumor sample were acquired using a ×4 Plan Fluor 0.13 NA Nikon objective on a TI-Eclipse TE2000-U inverted microscope equipped with a Nikon DS-Fi2 color camera and motorized Prior stage. DCIS structures were identified as intact non-disseminated tumor structures and were counted in the composite images. In addition, tumor area, which encompasses the whole tumor region, and cellular area, which consists of only the area with cancer cells (i.e., excluding the stroma), were measured; the cellular coefficient was calculated as cellular area/tumor area. The DCIS index was calculated as the product of the number of DCIS structures multiplied by the cellular coefficient (to correct for differences in cellularity among tumors). The DCIS index was then normalized over the tumor area (to correct for differences in tumor size).

**Interferometric photoactivated localization microscopy**. MCF7 cells, transduced with pCMV-mEos2-EVL-IRES-BlastR lentivirus, were cultured for 24 h on 25-mm-diameter coverslips containing gold nanoparticles (Nanopartz) that serve as iPALM calibration standards and drift correction fiducial markers[38,45]. Cells were fixed using 2% PFA in PEM buffer (80 mM PIPES, 5 mM EGTA, 2 mM MgCl₂, pH 6.8) and the actin cytoskeleton was stained using phalloidin conjugated to Alexa Fluor 647 (Thermo Fisher Scientific) at a concentration of 2.5% (5 U/mL). Immediately following labeling, samples were mounted in STORM buffer (100 mM MEA, 0.4 mg/mL glucose oxidase, and 24 μg/mL catalase from Sigma), 10% glucose, 10 mM NaCl, and 50 mM Tris-HCl, pH 8.0. A second 18 mm diameter coverslip was adhered to the top of the sample and sealed to create an air-tight imaging volume, permitting sequential 3D super-resolution imaging of Alexa Fluor 647-Phalloidin via STORM and mEos2-EVL via PALM. Samples were imaged using interferometric photoactivation and localization microscopy, as previously described. Briefly, images were acquired via two ×60 TIRF 1.49NA objectives (Nikon), with signals from both combined in a custom three-way beamsplitter (Rocky Mountain Instruments), and imaged in three phase channels on EMCCD cameras (iXon DU-897, Andor Technology). Alexa Fluor 647-conjugated phalloidin was imaged first, using 642 nm laser excitation (OptoEngine LLC) operating at ca. 5 kW/cm² at the sample, with camera exposure of 40 ms to acquire 60,000 raw frames. mEos2-EVL was imaged using 561 nm laser excitation (OptoEngine LLC) at ca. 2.5 kW/cm², with 2–10 W/cm² illumination at 405 nm (Coherent) for photoconversion of mEos2, and 50 ms camera exposure to acquire 50,000 raw images. Raw images were then subjected to iPALM localization, final image reconstruction, multichannel alignment, and analysis using PeakSelector software[46]. Final en face and orthogonal ROIs are displayed as cumulative intensity projections of rendered localizations.

**Laser scanning confocal microscopy**. Cells were embedded in a 3D type-I collagen matrix in 8-well glass-bottom chamber slides as previously described in Three-Dimensional Culture System methods. Cells were imaged using a ×60 Plan Apo 1.42 NA Olympus Objective on an Olympus Fluoview1200 laser scanning confocal microscope. Images are maximum intensity projections of confocal z-series.

**Fig. 6** EVL suppresses invasive activity of breast cancer cells and is associated with low dissemination in breast tumors. **a** Fold change in invasion of LKO and *EVL* KD control and E2-treated MCF7 cells; mean ± s.d.; *$p = 0.02$, ***$p = 0.004$ (unpaired *t* test). Data are from three independent experiments. **b** Fold change in invasion of control and tam-treated eGFP and eGFP-EVL MCF7 cells; mean ± s.d.; †$p = 0.04$, ††$p = 0.01$ (unpaired *t* test). Data are from three independent experiments. **c** DCIS index in LKO and constitutive *EVL* KD MCF7 xenograft tumors treated with E2 ($n = 3$ mice in LKO and $n = 5$ in *EVL* KD group; mean ± s.d. *$p = 0.04$) (Welch's *t* test). **d** DCIS index in control and inducible *EVL* KD tumors ($n = 7$ mice per group; mean ± s.d.; **$p = 0.001$) (Welch's *t* test). **e** Representative H&E stains (left), scale bar is 500 μm; and doxycycline induction (right), scale bar is 100 μm. **f** Representative luminal B breast tumors with high (top) or low (bottom) ER expression (Cedars-Sinai LumB TMA). Left panels show merged images of cytokeratin (red) and EVL (green) immunolabeling and middle panels of ER (red) and EVL (green) immunolabeling. Images are maximum intensity projections of z-series. Scale bar is 50 μm. Right panels are 3D reconstructions of boxed areas, imaged at higher resolution. Scale bar is 5 μm. **g** Quantification of EVL levels in ER low (1st quartile) and ER high (3rd quartile) tumors; a.u. = arbitrary units; mean ± s.e.m.; ****$p < 0.0001$ (Welch's *t* test). **h** Scatter plot of ER vs. EVL levels; bubble area is proportional to the number of positive lymph nodes in the corresponding patient; *r* is Pearson's correlation coefficient; correlation is significant at $p < 0.01$; a.u. = arbitrary units. **i** Quantification of EVL levels in LII low (≤7 μm) and LII high (>9 μm) tumors by mean fluorescence intensity; a.u. = arbitrary units; mean ± s.e.m. ****$p < 0.0001$ (Welch's *t* test). **j** Scatter plot of EVL levels and LII; bubble area is proportional to the number of positive lymph nodes in the corresponding patient; *r* is Pearson's correlation coefficient; correlation is significant at $p < 0.01$. **k** Survival of ER+ breast cancer patients clustered by *EVL* expression (split at median). Data from KM plotter (Affy ID = 217838_s_at/gene symbol = *EVL*; ER status derived from gene expression). Hazard ratio (HR) and *p* value shown in inset (log-rank test); 1541 patients in the *EVL* high group, 1541 patients in the *EVL* low group

**CRISPR/Cas9 knock-in**. Cells were edited using the CRISPR/Cas9 as previously described[47]. Briefly, the three following constructs were co-transfected to edit endogenous *EVL*: (1) a plasmid coding for the SpCas9 endonuclease; (2) a template plasmid coding for the sequence of eGFP and a flexible linker with an amino acid sequence of GGSGGSGGS flanked upstream and downstream by ~800 bp of the ATG codon of *EVL*; and (3) a linear double-stranded DNA product obtained by PCR amplification and purification coding for the U6 promoter, the genomic RNA (gRNA) targeting the ATG region of the *EVL* gene and the tracrRNA recognized by Cas9. The gRNA sequence was ACTTTTCAGCCATGGCCACA, the underlined nucleotides representing the start codon of the *EVL* gene. Edited cells (eGFP-

*EVL*edited) were generated by transfection, at 70% confluence, with 600 ng of each of the above-mentioned constructs using Fugene HD in OptiMEM. Edited cells were enriched by fluorescence-activated cell sorting on a FACSAria II and efficient editing of *EVL* was verified by fluorescence microscopy and validated by genomic PCR.

**Western blot analysis**. Cells were scraped on ice in lysis buffer (10% glycerol, 1% IGEPAL, 50 mM Tris, pH 7.5, 200 mM NaCl, 2 mM MgCl₂, protease inhibitor cocktail from Proteomics, M250-1ML, and phosphatase inhibitor cocktail from

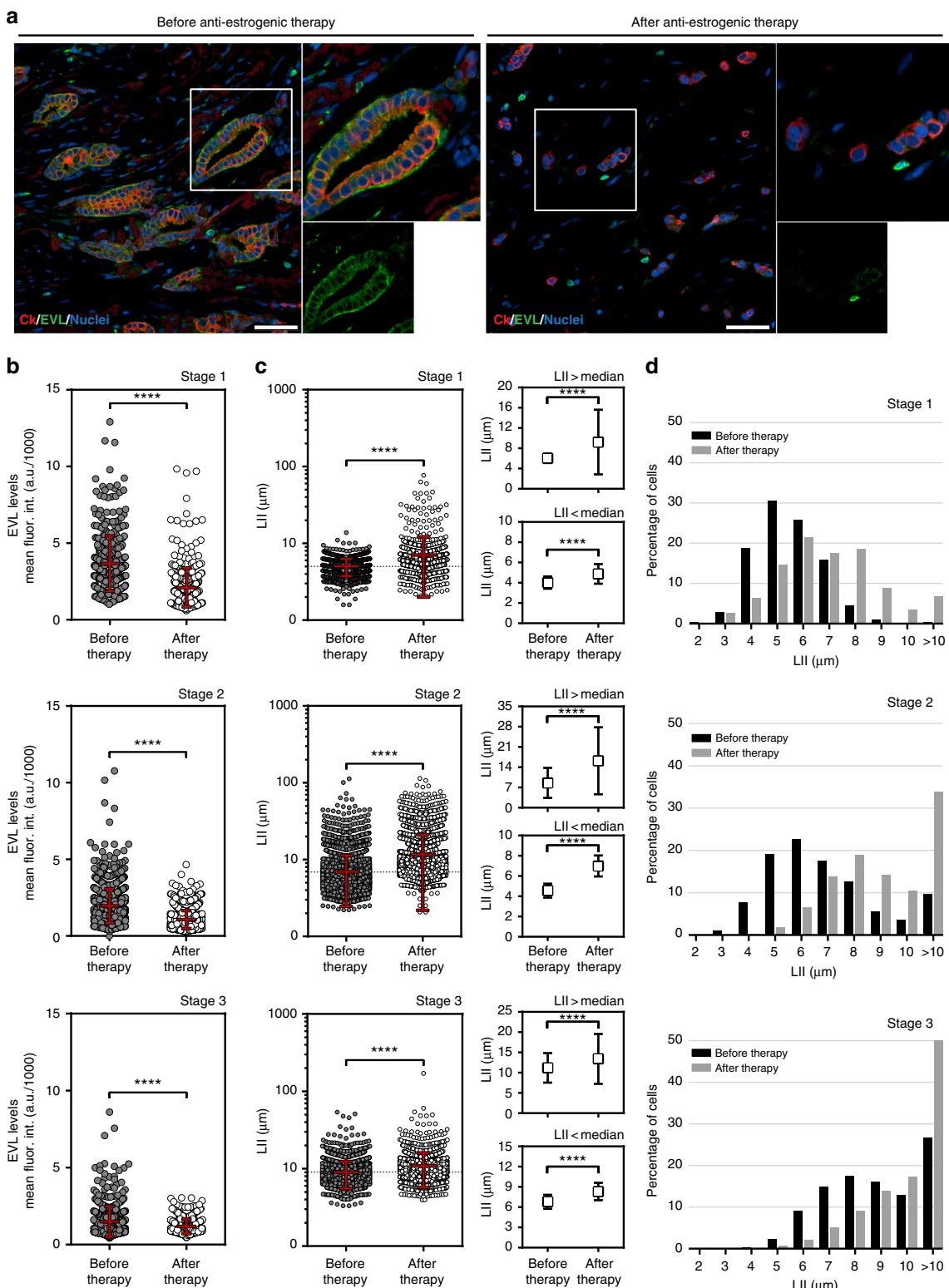

**Fig. 7** Anti-estrogenic hormone therapy is associated with decreased EVL expression and increased local invasion in tumors post-treatment. **a** Representative images from the same ER+ breast tumor before (left panel) and after (right panel) neoadjuvant hormone therapy. Human cytokeratin is shown in red, EVL in green, and nuclei in blue. Scale bar is 100 μm. In each panel, top-right inset shows magnification of the boxed area and bottom-right inset shows the EVL channel separately. **b** Quantification of EVL levels before and after hormone therapy. Scatter plots show the full range of cells analyzed within each tumor set: stage 1, $n = 425$ cells before and $n = 422$ cells after therapy; stage 2, $n = 1256$ cells before and $n = 1185$ cells after therapy; and stage 3, $n = 585$ before and $n = 501$ after therapy. Red lines represent mean ± s.d. ****$p < 0.0001$ (Welch's $t$ test). **c** Quantification of local invasion index (LII) before and after therapy. Scatter plots show the full range of cells analyzed within each tumor set: stage 1, $n = 1259$ cells before and $n = 1335$ cells after therapy; stage 2, $n = 7113$ cells before and $n = 3987$ cells after therapy; and stage 3, $n = 2701$ before and $n = 2644$ after therapy. Red lines represent mean ± s.d. ****$p < 0.0001$ (Welch's $t$ test); right insets show values from each scatter plot segregated into values > median and values < median (****$p < 0.0001$; Welch's $t$ test). **d** Histograms showing percentage of cells within different LII bins, before and after therapy

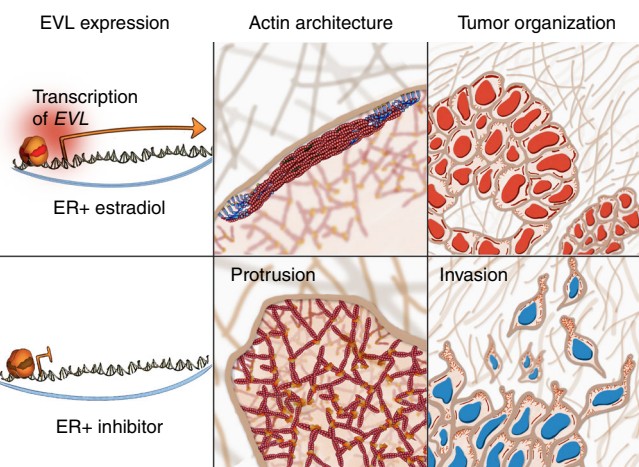

**Fig. 8** Model: Suppression of breast cancer cell invasion by ER. ER activates *EVL* transcription. EVL inhibits invasion by generating SCABs (suppressive cortical actin bundles), which suppress protrusive activity. ER inhibition leads to decrease in EVL levels and increase in protrusive activity and invasion. Red nuclei represent ER positivity

Boston BioProducts (BP479)). Lysates were incubated on ice for 15 min followed by clarification at 14,000 rpm for 15 min at 4 °C. Protein concentrations were measured using the Bradford assay (Thermo) and 25 μg total protein from each sample were resolved by SDS-polyacrylamide gel electrophoresis on 10% polyacrylamide gels. For film, samples were transferred onto polyvinylidene difluoride membranes (Millipore), which were blocked overnight in 5% BSA at 4 °C, and then probed with polyclonal rabbit anti-EVL antibody (a gift from Frank Gertler) at 1:2000 dilution, overnight at 4 °C, followed by standard western blotting protocol. Goat anti-rabbit or anti-mouse horseradish peroxidase secondary antibodies (Thermo Fisher) used with Enhanced Chemi-Luminescence (ECL) (BioRad Clarity Western ECL Substrate, 170-5061). To re-probe following ECL western blotting, antibodies were stripped in 0.2 M NaOH for 15 min at room temperature, and membranes were re-probed for actin, as a loading control, using monoclonal mouse anti-actin antibody (Abcam, ab3280, 1:1000) and developed following the same western blot protocol. For Licor, samples were transferred onto nitrocellulose membranes (GE Healthcare), which were blocked for 1 h in 0.5× casein (Thermo Fisher), and then probed with polyclonal rabbit anti-EVL antibody (Sigma Prestige HPA018849) at 1:500 dilution, overnight at 4 °C. Near-infrared fluorescent (Licor) secondary antibodies were used for detection. Membranes were scanned and re-probed with mouse anti-actin (ProteinTech 66009-1-Ig, 1:5000) and mouse anti-GFP (ProteinTech 50430-2-AP, 1:4000).

**Proliferation assay**. Cells were plated in a 6-well plate (25,000 cells per well) in growth media and allowed to adhere for 24 h. A full media change with drug was performed every day and maintained for 48 h, after which cells were trypsinized and quantified using a hemocytometer.

**Statistical analysis**. Unless indicated otherwise, statistical summaries are represented as mean ± s.e.m or mean ± s.d and statistical significance was determined using a two-tailed $t$ test with Welch's correction. The nonparametric Mann–Whitney test was used in the presence of skewed data. The Pearson's correlation coefficient was estimated to assess the relationship between continuous variables. For ChIP-qPCR experiments, $p$ values were generated by two-way ANOVA (analysis of variance) using the Sidak test for multiple comparison to one control. Survival was estimated using Kaplan–Meier plots. Comparison between survival curves was performed using the log-rank test.

**Data availability**. Survival data and RNA-seq data in the current study were derived by analysis of The Cancer Genome Atlas, a data repository publicly available at https://portal.gdc.cancer.gov. Microarray data are published and publicly available in the corresponding referenced studies. ChIP-seq data are deposited on GEO repository (GSE116768).

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

## Acknowledgements

We wish to acknowledge the Experimental Mouse Resource Service, particularly Gillian Paine-Murrieta and Bethany Skovan for their technical assistance; Agnes Witkiewicz and the Tissue Acquisition and Cellular/Molecular Analysis (TACMSR), particularly Betsy Dennison for procuring patient samples; Paul Krieg (University of Arizona, Tucson, AZ, USA) for comments on the manuscript; the staff at the Advanced Imaging Center at Janelia Research Campus for their assistance with iPALM imaging, a facility generously supported by the Gordon and Betty Moore Foundation and the Howard Hughes Medical Institute; Jonathan Kaye (Cedars-Sinai, Los Angeles, CA, USA) for the Cedars-Sinai LumB TMA and the National Cancer Institute Cancer Diagnosis Program for the BCP TMA. This research was supported by the NCI grant RO1 CA196885-01 (to G.M.), diversity supplemental RO1 CA196885-01 grant (to J.I.P.), Science Foundation Arizona Bisgrove Scholars Postdoctoral Fellowship (to S.S.P.), and the NCI University of Arizona Cancer Center Support Grant P30CA023074.

## Author contributions

G.M. and M.P.-R. designed the study and prepared the manuscript. M.P.-R., S.S.P., D.G.A., J.I.P., A.W.W., S.M.H., and J.A. performed and analyzed experiments. S.S.P., J.A., M.P.-R., and S.M.H. performed and analyzed the super-resolution imaging experiments at Janelia Research Campus. T.W. performed and analyzed the ChIP-seq and ChIP-PCR experiment. M.N. analyzed the TCGA data. R.G. generated the CRISPR cells. D.T. contributed to developing the LII. D.J.R. provided input on the statistical analyses. B.K. provided reagents, and contributed intellectually to the analysis and interpretation of the data. M.P.-R., S.S.P., T.W., J.I.P., J.A., B.K., and G.M. wrote the manuscript.

## Additional information

**Competing interests:** The authors declare no competing interests.

