## [Peer Review File · Nature Communications]

Reviewers' comments:

Reviewer #1 (Remarks to the Author):

Marco Padilla-Rodriguez et al demonstrated that ER suppresses invasion of ER+ breast cancer cells by promoting the generation of SCABs which inhibit leading edge motility dynamics. They further showed that ER activation transcriptionally upregulates EVL which promotes SCABs and suppresses invasion. In addition, the authors confirmed that hormone therapy of breast cancer patients resulted in suppression of EVL expression, which was associated with increased local invasion in tumors post treatment. Although this paper is well written and some of their findings are interesting, the conclusion is premature and required substantial evidence.

- (1) Though authors showed that ER directly binds to EVL genomic regions, more experiments are needed to demonstrate which site(s) play the most important role in activating EVL transcription.
- (2) The invasion assay used in this manuscript is different from the classic Boyden-chamber invasion assay, and requires 24-48 hr of incubation time. Thus cellular proliferation (or outgrowth of tumor cells) could contribute to the phenotype seen. Can these findings be repeated in Boyden-chamber invasion assay?
- (3) Do ER-negative breast cancer cell lines lost these phenotypes? ER-signaling has also been shown to promote breast tumor cell migration and invasion (via ERK, AKT, MMP, and etc), under what condition do these counterintuitive signaling achieve a promoting or inhibiting effect on migration and invasion?
- (4) Genomic location(s) should be added to Figure 4c to make the region presented clear to the reader. And it would be ideal if the primers used in Fig 4d could be indicated in Figure 4c.
- (5) The authors need to specify in which cell line they performed ChIP-seq and ChIP-qPCR assays.
- (6) There's a typo in Supplementary methods part describing ChIPseq: set $q < 10^{-200}$ as cutoff would be unpractical.
- (7) The authors should show the EVL level after ectopic expression of GFP-EVL in breast cancer cells.

Reviewer #2 (Remarks to the Author):

This publication describes a novel mechanism by which estrogen receptor signaling regulates cell invasion via expression of EVL, an actin regulator. More specifically, the data here demonstrate that estrogen suppresses invasion, in contrast to its well-described effect of inducing tumor growth. The authors show that suppression of ER signaling in patients treated with neo-adjuvant therapy led to decreased EVL and increased local invasion. They go on to show that ER signaling regulates EVL expression leading to changes in cortical actin organization, which affects protrusive ability and ultimately invasion. These findings are timely in the context of understanding how drugs currently used in the clinic affect proliferation vs. migration. Studies like this one are essential to better understand the effects of current treatments on different phenotypic behaviors associated with both tumor growth and metastasis. This study includes data from in vitro cells, in vivo mouse models and human patients, which strengthens the work. The authors use innovative approaches to quantify their data, particularly in the tumor sections, which provide a more complete assessment of the phenotypic diversity present in tumors.

Additional evidence could further help strengthen the conclusions:

SCAB formation: The authors describe a novel cytoskeletal feature described as Suppressive Cortical Actin Bundles or SCABs, rich in pMLC, which increase contractility of the cell and decrease protrusion formation and subsequent invasion. The authors show that these SCABs form in a variety of epithelial cells and regulated by ER.

- SCAB formation is restricted to the outside rim of the group of cells. It would be helpful for the authors to show images covering an entire clump of cells. Do the SCABs extend around the entire periphery continuously of these clumps? Are they only located on the outside rim of the cells? Does local protrusion lead to SCAB formation on the cells behind the leader cells once it has left?
- Do SCABs form in 3D? Can the authors visualize SCABs in their 3D invasion assay?
- Finally, can these be visualized in tumor sections? It would be helpful if the authors could stain for pMLC in their tumor sections, comparing pMLC levels in tumor cells close together, versus the ones that have started to invade as single cells.

ER-mediated EVL expression regulation: The authors show very nicely that ER signaling activated by estradiol directly regulates EVL mRNA expression (Fig 4). However, the authors only show changes in mRNA levels regulated by ER signaling. They should also show that this leads to significant changes in EVL protein levels, by Western Blot or IF for example.

Mechanism: What is the mechanism by which EVL regulates SCAB formation and pMLC? The authors briefly mention co-localization of EVL/pMLC in these cells (Fig 5C), however this is not evident from the image shown, and has not been quantified. Does the co-localization amount change with ER signaling? Does this occur through ROCK?

Tools for EVL level manipulation: the authors use an shRNA for EVL but only show 40% knockdown of the protein in Fig S4. Given the low level of knockdown, it would be important to use an additional shRNA targeting EVL. The authors also use an eGFP-EVL construct to overexpress EVL, however it is not clear what level of overexpression is induced relative to endogenous. A western blot is needed to demonstrate this.

Human Hormone Therapy data: The data in Figure 7 refers to patients treated with neo-adjuvant hormone therapy. Can the authors be more specific about this? This is confusing as the authors mention HRT in the beginning, and it is not clear what hormone therapy is here. Likely this is with an ER antagonist but this needs to be explained for clarity.

The data is shown as number of cells quantified, but how many patients are these from? Did these patients exhibit a clinical response to the hormone therapy ie. by RECIST or any other criteria?

Statistical methods and ability to reproduce data:

Overall, the methods are very detailed and statistical analysis is well done. Two issues:

- There are a few panels however where the authors show a representative graph for one experiment, without error bars or statistics Fig 1l, Fig 4 e/f, Fig 5f/g. It is hard to gauge the reproducibility and significance of this data the way it is presented.
- It is not clear from the methods where the samples used in Fig 7 were obtained from. Unless, as these from one of the TMAs? Please clarify.

Response to the reviewers: We would like to thank the reviewers for their assessment of our study, and provide a point-by-point response to their comments.

Reviewer #1 (Remarks to the Author):

Marco Padilla-Rodriguez et al demonstrated that ER suppresses invasion of ER+ breast cancer cells by promoting the generation of SCABs which inhibit leading edge motility dynamics. They further showed that ER activation transcriptionally upregulates EVL which promotes SCABs and suppresses invasion. In addition, the authors confirmed that hormone therapy of breast cancer patients resulted in suppression of EVL expression, which was associated with increased local invasion in tumors post treatment. Although this paper is well written and some of their findings are interesting, the conclusion is premature and required substantial evidence.

We thank the reviewer for their comments on our study. We think that the new additions to the manuscript will address several of their concerns and further substantiate our conclusion.

(1) Though authors showed that ER directly binds to EVL genomic regions, more experiments are needed to demonstrate which site(s) play the most important role in activating EVL transcription.

In our ChIP experiments, we have identified 12 high-confidence ER binding sites in and around the *EVL* gene. These experiments only prove that the binding is of high-confidence, but do not determine which of these sites are most important for ER-promoted transcription, as there is no linear correlation between binding confidence and transcription. In addition, these sites are located far from the *EVL* transcription initiation site and could be functioning, in various combinations, as enhancers, thus precluding us from designing a simple reporter assay to determine which site plays the most important role in driving *EVL* transcription. Therefore, to be able to examine the significance of each of these sites, separately and in combination, one would need to use genomic editing to modulate the binding of ER at these sites in their native chromatin structure, and assess the potential corresponding functional outcomes. Performing such experiments would be an onerous task (multiple papers worth of work).

We do agree with the reviewer that characterizing these sites would be beneficial for better understanding the biological mechanism of how ER is regulating actin cytoskeletal remodeling. However, considering that these sites do not harbor any mutations that are commonly observed in aggressive tumors, we think that ranking their significance would not add to the impact of this study, particularly at the translational level.

(2) The invasion assay used in this manuscript is different from the classic Boyden-chamber invasion assay, and requires 24-48 hr of incubation time. Thus cellular proliferation (or outgrowth of tumor cells) could contribute to the phenotype seen. Can these findings be repeated in Boyden-chamber invasion assay?

Based on our experience in studying cell migration and invasion, we believe that although suitable for studying the invasive behavior of aggressive cells, like triple-negative breast cancer cells, classic Boyden-chamber invasion assays are not suitable to examine invasion of less invasive cells, like ER-positive breast cancer cells. In Boyden-chamber invasion assays, the cells need to go through a confined space (determined by the pore size and thickness of the transwell filters), in addition to transitioning from 2D to 3D migration, and having to degrade a relatively thick layer of Matrigel (orders of magnitude thicker than a physiologically relevant basement membrane). These factors directly alter the mode of migration, and introduce variables not immediately relevant to the biological question. To overcome these obstacles, we have developed and optimized the invasion assay we present in this study. In our assay, the cells are plated in a 3D microenvironment that is directly connected to the surrounding matrix; therefore, the cells are not faced with a physical barrier and can readily invade the surrounding matrix, with which they are in direct contact. We think that our assay is more permissive in allowing cells to invade and is thus more suitable to quantify invasion of relatively less aggressive cancer cells.

Regarding the reviewer's concern about the confounding factor of cell proliferation, since estrogen treatment is expected to enhance proliferation and ER inhibition to suppress it, a change in proliferation would not exaggerate our results but in fact would attenuate them since we are showing that estrogen treatment suppresses invasion and ER inhibition promotes it. We confirmed these changes in cell proliferation at 48 hours after treatment with estrogen and ER inhibitors. We added these results to the manuscript in Supplemental Fig.S1h.

(3) Do ER-negative breast cancer cell lines lose these phenotypes? ER-signaling has also been shown to promote breast tumor cell migration and invasion (via ERK, AKT, MMP, and etc), under what condition do these counterintuitive signaling achieve a promoting or inhibiting effect on migration and invasion?

These actin remodeling phenotypes are less prominent in ER-negative breast cancer cells, and increasing the expression of EVL significantly curbs their invasive behavior. We tested this in the triple-negative breast cancer cell line SUM159 and found that overexpression of EVL suppressed their invasive activity in our 3D invasion assay. We have added these results to the manuscript in Supplemental Fig.S7a.

On the other hand, we have no evidence suggesting that the signaling pathways mentioned by the reviewer suppress invasion under these particular conditions. Rather, we think that suppression of invasion by SCABs could be independent from these pathways and could play a dominant role over other actin remodeling programs mediated by these pathways. Investigating these concepts requires understanding the mechanisms that drive invasion in the absence of SCABs (when EVL levels are low), which is within our future plans but, we believe, is beyond the scope of this study.

(4) Genomic location(s) should be added to Figure 4c to make the region presented clear to the reader. And it would be ideal if the primers used in Fig 4d could be indicated in Figure 4c.

We have added the suggested genomic coordinates in Fig.4c. In addition, we added a track in Fig.4c showing the amplicons of the PCR reactions presented in Fig.4d. Visually indicating the primers

proved impractical at the scale of the resolution of the figure as they would seemingly overlap. For clarity, we have updated the supplementary table containing the primer sequences with better description, full genomic coordinates, and sequences. The table now is organized as Peaks, Amplicons/Targets, and oligos. We appreciate the constructive suggestions, and hope the readability of Fig.4 is now better.

(5) The authors need to specify in which cell line they performed ChIP-seq and ChIP-qPCR assays.

All ChIP-seq data was generated from experiments done in MCF7 cells. Putative enhancer binding sites (as called by MACS2) in and upstream of the EVL gene were selected for ChIP-qPCR validation in independent experiments. We accordingly labeled the corresponding panels in Fig.4.

(6) There's a typo in Supplementary methods part describing ChIPseq: set $q < 10^{-200}$ as cutoff would be unpractical.

We verified that the q-value reported in the methods ($q < 10^{-200}$) is the correct value from the MACS2 peak caller (we double-checked it in our bioinformatics pipeline). We appreciate that it is a very stringent cutoff; however, the peaks we have reported are within the top ten percentile of peaks based on the q-values generated by MACS2. The large number of peak reads in our ChIP sample compared to input DNA corresponds to very high statistical significance in MACS2 (the MACS2 q-value is a reflection of the number of reads per million contributing to the peaks in the dataset, in the ChIP sample versus the input sample). In our experience, such a q-value does predict real binding quite well.

(7) The authors should show the EVL level after ectopic expression of GFP-EVL in breast cancer cells.

We have added western blot analysis showing the levels of EVL ectopic expression versus endogenous expression in Supplementary Fig.S5a.

Reviewer #2 (Remarks to the Author):

This publication describes a novel mechanism by which estrogen receptor signaling regulates cell invasion via expression of EVL, an actin regulator. More specifically, the data here demonstrate that estrogen suppresses invasion, in contrast to its well-described effect of inducing tumor growth. The authors show that suppression of ER signaling in patients treated with neo-adjuvant therapy led to decreased EVL and increased local invasion. They go on to show that ER signaling regulates EVL expression leading to changes in cortical actin organization, which affects protrusive ability and ultimately invasion. These findings are timely in the context of understanding how drugs currently used in the clinic affect proliferation vs. migration. Studies like this one are essential to better understand the effects of current treatments on different phenotypic behaviors associated with both tumor growth and metastasis. This study includes data from in vitro cells, in vivo mouse models and human patients, which strengthens the work. The authors use innovative approaches to quantify their data, particularly in the tumor sections, which provide a more complete assessment of the phenotypic diversity present in tumors.

We thank the reviewer for their comments on our study, particularly for highlighting the significance of the work. We have addressed most of the reviewer's suggestions and included a point-by-point response to their comments below.

Additional evidence could further help strengthen the conclusions:

SCAB formation: The authors describe a novel cytoskeletal feature described as Suppressive Cortical Actin Bundles or SCABs, rich in pMLC, which increase contractility of the cell and decrease protrusion formation and subsequent invasion. The authors show that these SCABs form in a variety of epithelial cells and regulated by ER.

(1) SCAB formation is restricted to the outside rim of the group of cells. It would be helpful for the authors to show images covering an entire clump of cells. Do the SCABs extend around the entire periphery continuously of these clumps? Are they only located on the outside rim of the cells? Does local protrusion lead to SCAB formation on the cells behind the leader cells once it has left?

SCABs are localized around the periphery of cell clusters; they are mostly present at the outside rim, and coincide with the absence of protrusions. However, since ER-positive cells are epithelial in their characteristics, we have not observed SCABs in the context of cell dissemination. Large stitched images shown in Supplementary Fig.S5b support this observation.

(2) Do SCABs form in 3D? Can the authors visualize SCABs in their 3D invasion assay?

We expressed eGFP-EVL and MLC-mRuby2 in MCF7 cells to visualize SCABs in 3D collagen matrix equivalent to the invasion assay matrix. Imaging 3D clusters using laser-scanning confocal microscopy revealed cortical co-localization of EVL and MLC. We have added these images in Supplementary Fig.S6c.

(3) Finally, can these be visualized in tumor sections? It would be helpful if the authors could stain for pMLC in their tumor sections, comparing pMLC levels in tumor cells close together, versus the ones that have started to invade as single cells.

Immunofluorescent labeling of phospho-MLC is not compatible with formalin-fixation paraffin-embedding of tissue, which is standard procedure for processing tissue from clinical samples. Therefore, we used EVL as a marker for SCABs in tissue samples, which exhibited a localization at the outside rim of cell clusters (similarly to the *in vitro* data). We added more images taken from six different tumors in Supplemental Fig.S7c. In addition, our data in Fig.6f show that SCABs are particularly absent in tumors that disseminate as single cells.

(4) ER-mediated EVL expression regulation: The authors show very nicely that ER signaling activated by estradiol directly regulates EVL mRNA expression (Fig 4). However, the authors only show changes in mRNA levels regulated by ER signaling. They should also show that this leads to significant changes in EVL protein levels, by Western Blot or IF for example.

We performed immunofluorescent labeling of endogenous EVL in cells treated with estrogen or ER inhibitors. Analysis of the staining revealed that, in agreement with the qPCR data, EVL levels are higher in estrogen-treated cells and lower in cells treated with ER inhibitors. We have added representative images to Fig.4f. and the corresponding quantification to Fig.4g.

(5) Mechanism: What is the mechanism by which EVL regulates SCAB formation and pMLC? The authors briefly mention co-localization of EVL/pMLC in these cells (Fig 5C), however this is not evident from the image shown, and has not be quantified. Does the co-localization amount change with ER signaling? Does this occur through ROCK?

EVL protein structure does not predict direct binding to myosin. Therefore, we do not believe that the mechanism by which EVL regulates SCABs is dependent on direct interaction between EVL and MLC. Rather, EVL promotes the generation of actin filaments that are bundled into contractile structures at the membrane. We think the localization of EVL to SCABs is dependent on ROCK in that the inhibition of ROCK ultimately abrogates the SCABs by suppressing contractility; however, we have no data suggesting that ROCK activity is specifically promoting the localization. We have edited the text to focus the narrative on the localization of EVL to the SCABs. In addition, in the revised version of the manuscript, we included super-resolution microscopy analysis of the localization of EVL at SCABs, which better illustrates the enrichment of EVL at cortical bundles and at focal adhesions. This analysis is included in Fig.5e.

(6) Tools for EVL level manipulation: the authors use an shRNA for EVL but only show 40% knockdown of the protein in Fig S4. Given the low level of knockdown, it would be important to use an additional shRNA targeting EVL. The authors also use an eGFP-EVL construct to overexpress EVL, however it is not clear what level of overexpression is induced relative to endogenous. A western Blot is needed to demonstrate this.

As mentioned above, we show that EVL localizes at focal adhesions in this study. Also, we have recently completed a study examining the role of EVL at focal adhesion (if necessary, we would be

glad to share the manuscript with the reviewer and the editors at Nature Communications). In both studies we found that a relatively high level of EVL knockdown results in extremely weakened adhesions, to the degree that cells with high knockdown detach in culture, and over time only cells with relatively low knockdown remain. We have included here images of EVL knockdown cells with such weakened adhesions. For that reason, we have limited our studies to lower knockdown cells. Importantly, we have validated the shRNA used here previously (Mouneimne et al 2012) and in our new study.

(7) Human Hormone Therapy data: The data in Figure 7 refers to patients treated with neo-adjuvant hormone therapy. Can the authors be more specific about this? This confusing as the authors mention HRT in the beginning, and it is not clear what hormone therapy is here. Likely this is with an ER antagonist but this needs to be explained for clarity. The data is shown as number of cells quantified, but how many patients are these from? Did these patients exhibit a clinical response to the hormone therapy ie. by RECIST or any other criteria?

We have clarified this point by indicating in the text that the hormone therapy is an anti-estrogenic therapy, which is (as the reviewer pointed out) distinct from the HRT mentioned in the beginning of the paper. In addition, we included the exact treatments of each patient, along all the clinical information we obtained, in the supplementary methods. Unfortunately, we do not have information on clinical response to the hormone therapy. Each set of samples (before and after therapy) is from one patient; this is better clarified in the methods now.

(8) Statistical methods and ability to reproduce data:

Overall, the methods are very detailed and statistical analysis is well done. Two issues:

- There a few panels however where the authors show a representative graph for one experiment, without error bars or statistics Fig 1l, Fig 4 e/f, Fig 5f/g. It is hard to gage the reproducibility and significance of this data the way it is presented.
- It is not clear from the methods where the samples using in Fig 7 were obtained from. Unless, as these from one of the TMAs? Please clarify.

We have re-analyzed the experiments that the reviewer pointed out and expressed the data as fold change, averaged the data from different repeats, and presented stats on the plots. These changes can be seen in the following figures: Fig.1l, Fig.4e, Fig.6a, Fig.6b and Supplemental Fig.S3c.

Samples shown in Fig.7a and Supplemental Fig.S8a are from patients. These figures have now been more clearly labeled to better communicate their origin.

REVIEWERS' COMMENTS:

Reviewer #1 (Remarks to the Author):

The authors have carefully addressed the comments raised previously and revised the manuscript accordingly, no further comment on this manuscript.

Reviewer #2 (Remarks to the Author):

The Authors have addressed all the comments.